# BEHAVIOR PRIOR REPRESENTATION LEARNING FOR OFFLINE REINFORCEMENT LEARNING

**Hongyu Zang**[1], **Xin Li**[1],[*] **Jie Yu**[1], **Chen Liu**[1], **Riahsat Islam**[2],
**Rémi Tachet des Combes**[†], **Romain Laroche**[†]

[1] Beijing Institute of Technology, China      [2] Mila, Quebec AI Instiute, Canada
`{zanghyu,xinli,yujie,chenliu}@bit.edu.cn`
`riashat.islam@mail.mcgill.ca`   `{remi.tachet,romain.laroche}@gmail.com`

## ABSTRACT

Offline reinforcement learning (RL) struggles in environments with rich and noisy inputs, where the agent only has access to a fixed dataset without environment interactions. Past works have proposed common workarounds based on the pre-training of state representations, followed by policy training. In this work, we introduce a simple, yet effective approach for learning state representations. Our method, Behavior Prior Representation (BPR), learns state representations with an easy-to-integrate objective based on behavior cloning of the dataset: we first learn a state representation by mimicking actions from the dataset, and then train a policy on top of the fixed representation, using any off-the-shelf Offline RL algorithm. Theoretically, we prove that BPR carries out performance guarantees when integrated into algorithms that have either policy improvement guarantees (conservative algorithms) or produce lower bounds of the policy values (pessimistic algorithms). Empirically, we show that BPR combined with existing state-of-the-art Offline RL algorithms leads to significant improvements across several offline control benchmarks. The code is available at https://github.com/bit1029public/offline_bpr.

## 1 INTRODUCTION

Offline Reinforcement Learning (Offline RL) is one of the most promising data-driven ways of optimizing sequential decision-making. Offline RL differs from the typical settings of Deep Reinforcement Learning (DRL) in that the agent is trained on a fixed dataset that was previously collected by some arbitrary process, and does not interact with the environment during learning (Lange et al., 2012; Levine et al., 2020). Consequently, it benefits the scenarios where online exploration is challenging and/or unsafe, especially for application domains such as healthcare (Wang et al., 2018; Gottesman et al., 2019; Satija et al., 2021) and autonomous driving (Bojarski et al., 2016; Yurtsever et al., 2020). A common baseline of Offline RL is *Behavior Cloning* (BC) (Pomerleau, 1991). BC performs maximum-likelihood training on a collected set of demonstrations, essentially mimicking the behavior policy to produce predictions (actions) conditioned on observations. While BC can only achieve proficient policies when dealing with expert demonstrations, Offline RL goes beyond the goal of simply imitating and aims to train a policy that improves over the behavior one. Despite promising results, Offline RL algorithms still suffer from two main issues: i) difficulty dealing with limited high-dimensional data, especially visual observations with continuous action space (Lu et al., 2022); ii) implicit under-parameterization of value networks exacerbated by highly re-used data, that is, an expressive value network implicitly behaves as an under-parameterized one when trained using bootstrapping (Kumar et al., 2021a;b).

In this paper, we focus on state representation learning for Offline RL to mitigate the above issues: projecting the high-dimensional observations to a low-dimensional space can lead to a better performance given limited data in the Offline RL scenario. Moreover, disentangling representation learning from policy training (or value function learning), referred to as pre-training the state representations, can potentially mitigate the "implicit under-parameterization" phenomenon associated with the emergence of low-rank features in the value network (Wang et al., 2022). In contrast to

---

[*]Correspondence to Xin Li. This work was partially supported by NSFC under Grant 62276024 and 92270125.

[†]Work done while at Microsoft Research Montreal.

previous work that pre-train state representations by specifying the required properties, *e.g.*, maximizing the diversity of states encountered by the agent (Liu & Abbeel, 2021; Eysenbach et al., 2019), exploring the attentive knowledge on sub-trajectory (Yang & Nachum, 2021), or capturing temporal information about the environment (Schwarzer et al., 2021a), we consider using the behavior policy to learn generic state representations instead of specifying specific properties.

Many existing Offline RL methods regularize the policy to be close to the behavior policy (Fujimoto et al., 2019; Laroche et al., 2019b; Kumar et al., 2019) or constrain the learned value function of OOD actions not to be overestimated (Kumar et al., 2020; Kostrikov et al., 2021). Beyond these use, the behavior policy is often ignored, as it does not directly provide information on the environment. However, the choice of behavior has a huge impact on the Offline RL task. As shown by recent theoretical work (Xiao et al., 2022; Foster et al., 2022), under an agnostic baseline, the Offline RL task is intractable (near optimality is exponential in the state space size), but it becomes tractable with a well-designed behavior (*e.g.* the optimal policy or a policy trained online). This impact indicates that the information collected from the behavior policy might deserve more attention.

To this end, we propose Behavior Prior Representation (BPR), a state representation learning method tailored to Offline RL settings (Figure 1). BPR learns state representations implicitly by enforcing them to be predictive of the action performed by the behavior policy, normalized to be on the unit sphere. Then, the learned encoder is frozen and utilized to train a downstream policy with any Offline RL algorithms. Intuitively, to be predictive of the normalized actions, BPR encourages the encoder to ignore the task-irrelevant information while maintaining the task-specific knowledge relative to the behavior, which we posit is efficient for learning a state representation. Theoretically, we prove that BPR carries out performance guarantees when combined with conservative or pes-

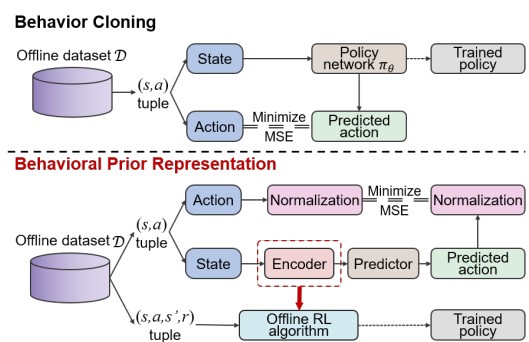

Figure 1: Illustration of Behavior Prior Representations and comparison with Behavior Cloning.

simistic Offline RL algorithms. While an uninformative behavioral policy may lead to bad representations and therefore degraded performance, we note that such a scenario may be predicted from the empirical returns of the dataset. Furthermore, since the learning procedure of BPR does not involve value functions or bootstrapping methods like Temporal-Difference, it can naturally mitigate the "implicit under-parameterization" phenomenon. We prove this empirically by utilizing effective dimensions measurement to evaluate feature compactness in the value network's penultimate layer. The key contributions of our work are summarized below:

- We propose a simple, yet effective method for state representation learning in Offline RL, relying on the behavior cloning of actions; and find that this approach is effective across several offline benchmarks, including raw state and pixel-based ones. Our approach can be combined to any existing Offline RL pipeline with minimal changes.

- Behavior prior representation (BPR) is theoretically grounded: we show, under usual assumptions, that policy improvement guarantees from offline RL algorithms are retained through the BPR, at the only expense of an additive behavior cloning error term.

- We provide extensive empirical studies, comparing BPR to several state representation objectives for Offline RL, and show that it outperforms the baselines across a wide range of tasks.

## 2 RELATED WORK

**Offline RL with behavior regularization.** Although to the best of our knowledge, we are the first to leverage behavior cloning (BC) to learn a state representation in Offline RL, we remark that combining Offline RL with behavior regularization has been considered previously by many works. A common way of combining BC with RL is to utilize it as a reference for policy optimization with baseline methods, such as natural policy gradient (Rajeswaran et al., 2018), DDPG (Nair et al., 2018; Goecks et al., 2020), BCQ (Fujimoto et al., 2019), SPIBB (Laroche et al., 2019a; Nadjahi et al.,

2019; Simão et al., 2020; Satija et al., 2021; Brandfonbrener et al., 2022), CQL (Kumar et al., 2020), and TD3 (Fujimoto & Gu, 2021). Other previous works include learning adaptive behavior policies that are biased towards higher-rewards trajectories (Cang et al., 2021), and pretraining behavior policies to guide downstream policies training (Zhang et al., 2021b). However, all these approaches neglect the potential of using the behavior policy to guide representation learning. In contrast, we investigate state representation learning via a BC-style approach and show that the downstream policy can be greatly boosted in that regime.

**Representation learning in Offline RL.** Pretraining representation has been recently studied in Offline RL settings, where several studies presented its effectiveness (Arora et al., 2020; Schwarzer et al., 2021a; Nachum & Yang, 2021a). Some typical auxiliary tasks for pretraining state representations include capturing the dynamical (Nachum & Yang, 2021b) and temporal (Schwarzer et al., 2021a) information of the environment, exploring the attentive knowledge on sub-trajectory (Yang & Nachum, 2021), improving policy performance by applying data augmentations techniques to the pixel-based inputs (Chen et al., 2021; Lu et al., 2022)... While driven by various motivations, most of these methods can be thought of as including different inductive biases via the specific design of the target properties of the representation. In this paper, the inductive bias we enforce is that the representation should allow matching the behavior action normalized to the unit hypersphere, we demonstrate its effectiveness on both raw-state and visual-observation inputs tasks. Additional related works are discussed in appendix B.

## 3 PRELIMINARIES

**Offline RL** We consider the standard Markov decision process (MDP) framework, in which the environment is given by a tuple $\mathcal{M} = (\mathcal{S}, \mathcal{A}, T, \rho, r, \gamma)$, with state space $\mathcal{S}$, action space $\mathcal{A}$, transition function $T$ that decides the next state $s' \sim T(\cdot|s, a)$, initial state distribution $\rho$, reward function $r(s, a)$ bounded by $R_{\max}$, and a discount factor $\gamma \in [0, 1)$. The agent in state $s \in \mathcal{S}$ selects an action $a \in \mathcal{A}$ according to its policy, mapping states to a probability distribution over actions: $a \sim \pi(\cdot|s)$. We make use of the state value function $V^\pi(s) = \mathbb{E}_{\mathcal{M},\pi} \left[ \sum_{t=0}^\infty \gamma^t r(s_t, a_t) \mid s_0 = s \right]$ to describe the long term discounted reward of policy $\pi$ starting at state $s$. In Offline RL, we are given a fixed dataset of environment interactions that include $N$ transition samples, i.e., $\mathcal{D} = \{s_i, a_i, s'_i, r_i\}_{i=1}^N$. We assume that the dataset $\mathcal{D}$ is generated i.i.d. from a distribution $\mu(s, a)$ that specifies the effective behavior policy $\pi_\beta(a|s) = \mu(s, a) / \sum_a \mu(s, a)$, and denote by $d^{\pi_\beta}(s)$ its discounted state occupancy density. Following Rashidinejad et al. (2021), the goal of Offline RL is to minimize the suboptimality of $\pi$ with respect to the optimal policy $\pi^\star$ given a dataset $\mathcal{D}$:

$$\text{SubOpt}(\pi) = \mathbb{E}_{\mathcal{D} \sim \mu} \left[ \mathcal{J}(\pi^\star) - \mathcal{J}(\pi) \right] = \mathbb{E}_{\mathcal{D} \sim \mu} \left[ \mathbb{E}_{\mathbf{s}_0 \sim \rho} \left[ V^\star(\mathbf{s}_0) - V^\pi(\mathbf{s}_0) \right] \right], \quad (1)$$

where $\mathcal{J}(\pi) = \mathbb{E}_{\mathbf{s}_0 \sim \rho} \left[ V^\pi(\mathbf{s}_0) \right]$ is the performance of the policy $\pi$.

**State Representation Learning** In this paper, our objective is to find a state representation function $\phi : \mathcal{S} \to \mathcal{Z}$ that maps each state $s \in \mathcal{S}$ to its representation $z = \phi(s) \in \mathcal{Z}$. The desired representations should provide necessary and useful information to summarize the task-relevant knowledge and facilitate policy learning. To investigate the capacity of state representation learning for mitigating the "implicit under-parameterization" phenomenon, following Lyle et al. (2022) and Kumar et al. (2021a), we measure the compactness of the feature in the penultimate layer of the value network by utilizing the *Effective Dimension*, where we refer to the output of the penultimate layer of the state-action value network as the feature matrix $\Psi \in \mathbb{R}^{|\mathcal{S}||\mathcal{A}| \times d}$ [1]:

**Definition 1.** *Effective Dimension Let* $\text{ED}(M)$ *denote the eigenvalues of a square matrix $M$, $\mathcal{D}$ be the offline dataset, and $\epsilon$ be a fixed hyperparameter. Then, the effective dimension of $\Psi$ is defined as*

$$\zeta(\Psi, \mathcal{D}, \epsilon) = \mathbb{E}_{\mathcal{D}} \left[ \left| \left\{ \sigma \in ED \left( |\mathcal{S}|^{-1} |\mathcal{A}|^{-1} \Psi^\top \Psi \right) \mid \sigma > \epsilon \right\} \right| \right], \quad (2)$$

It is the expected number of eigenvalues of $\Psi^\top \Psi$ that are larger than $|\mathcal{S}||\mathcal{A}|\epsilon$. By studying this quantity, we can explicitly observe the usefulness of the state representation objective in mitigating the "implicit under-parameterization" problem in value networks (see Section 6.3 for the results).

---

[1] Notably, when we characterize the state by its corresponding state representation, the dimension of the feature matrix changes accordingly as $\mathbb{R}^{|\mathcal{Z}||\mathcal{A}| \times d}$. We defer a detailed discussion to Appendix D.

## 4 BEHAVIOR PRIOR REPRESENTATION

Given an offline dataset $\mathcal{D}$ consisting of $(s, a)$ pairs, our goal is to learn an encoder $\phi(s)$ that produces state representations $z = \phi(s)$ allowing efficient and successful downstream policy learning. During pre-training, the BPR network is comprised of two connected components: an *encoder* $\phi_\theta$ and a *predictor* $f_\omega$. The encoder maps the state to the representation space, while the predictor projects the representations onto the unit sphere in dimension $|\mathcal{A}|$. In state $s$, the BPR network outputs a *representation* $z_\theta = \phi_\theta(s)$, and a *prediction* $y = f_\omega(z_\theta)$. We then $\ell_2$-normalize the prediction $y$ and the action $a$ to $\overline{y} = y/\|y\|_2$ and $\overline{a} = a/\|a\|_2$ and minimize the mean squared error to train the state representation (or equivalently maximize the cosine similarity between $y$ and $a$):

$$\mathcal{L}_{\theta,\omega} = \mathbb{E}_{(s,a)\sim\mathcal{D}}\left[\|\overline{y} - \overline{a}\|_2^2\right] = \mathbb{E}_{(s,a)\sim\mathcal{D}}\left[2 - 2\cdot\frac{\langle f_\omega(\phi_\theta(s)), a\rangle}{\|f_\omega(\phi_\theta(s))\|_2 \cdot \|a\|_2}\right], \tag{3}$$

where we set the action from the pair $(s, a)$ as the target. During the training process, stochastic optimization is performed to minimize $\mathcal{L}_{\theta,\omega}$ with respect to $\theta$ and $\omega$. Note that although the encoder and predictor are updated together through this optimization procedure, **only the encoder $\phi_\theta$ is used in the downstream task**. At the end of the training, we keep the encoder $\phi_\theta$ fixed and build the downstream Offline RL agents on top of the state representation that BPR learnt.

**Implementation details** BPR method does not rely on any specific architecture as its encoder network. Depending on the nature of given inputs, it can either be a convolutional neural network (CNN) for visual observation inputs, or a multi-layer perceptron (MLP) for physically meaningful state inputs. The representation $z_\theta$ that corresponds to the output of the encoder is then projected to the action space. In this paper, the projection is done by an MLP consisting of two linear layers followed by rectified linear units (ReLU) (Nair & Hinton, 2010), and a final linear layer followed by a tanh activation layer.

## 5 THEORETICAL ANALYSIS

For simplicity, we consider the optimization problem of BPR without the normalization term as:

$$\min \frac{1}{n}\sum_{i=1}^{n} \|f(\phi(s_i)) - a_i\|_2^2 : \phi \in \Phi, f \in \mathcal{F}, \tag{4}$$

where $(s_i, a_i)$ is an i.i.d. sample from the offline dataset (of size $n$). BPR consists in learning a representation $\phi$ impacting the policy search. In other words, with BPR, the function class for the policy consists in $\Pi_{\text{BPR}} \doteq \{\pi, \text{s.t. } \exists\,\omega \text{ with } \pi(\cdot|s) = f_\omega(\phi_\theta(s))\ \forall s\}$, where $f_\omega$ is the neural model parameterized with $\omega$ characterizing the policy on top of embedding $\phi_\theta(\cdot)$. Like any representation learning technique, the potential benefits are (i) enhancing the signal-to-noise ratio and (ii) reducing the size of the policy search. In this section, we develop an analysis showing that the potential harm of using BPR is upper bounded by an error $\epsilon_\beta$ that we control in Section 5.3. To our knowledge, this is the first representation learning technique for Offline RL with such guarantees.

**Idealized assumptions :** Letting $\hat{x}$ denote an estimate of a quantity $x$ computed using $\mathcal{D}$, we start by considering the following idealized assumptions:

**Assumption 1.** *(1.1) Access to the true behavior:* $\pi_{\hat{\beta}} = \pi_\beta$. *(1.2) Access to the true performance of policies:* $\hat{\mathcal{J}}(\pi) = \mathcal{J}(\pi)$ *for all policies* $\pi$. *(1.3) The embedding $\phi$ allows to represent the behavior policy estimate:* $\pi_{\hat{\beta}}(a|\phi(s)) = \pi_{\hat{\beta}}(a|s) \in \Pi_{\text{BPR}}$. *(1.4) The Offline RL algorithm performs perfect optimization on top of $\phi$:* $\mathcal{J}_{\text{BPR}} = \max_{\pi \in \Pi_{\text{BPR}}} \hat{\mathcal{J}}(\pi)$.

**Theorem 1.** *Under idealized Assumption 1, BPR returns a policy that improves over the behavior policy:* $\mathcal{J}_{\text{BPR}} \geq \mathcal{J}(\pi_\beta)$.

Its proof[2] is immediate: under perfect estimation and optimization, the fact that $\beta \in \Pi_{\text{BPR}}$ guarantees policy improvement. The above assumptions are stringent, and we propose to relax them in two different ways: for conservative algorithms that derive safe policy improvement guarantees (Petrik et al., 2016; Fujimoto et al., 2019; Laroche et al., 2019b; Simão et al., 2020), and for pessimistic

---

[2]For readability, we defer rigorous proofs of theorems to the Appendix E.

algorithms that derive a value function lower bound (Kumar et al., 2020; Buckman et al., 2021). We note that the policy improvement is constrained by the set $\Pi_{\text{BPR}}$ over which the policy search is conducted. Appendix G.6 empirically shows that the method will fail to improve over the behavioral policy in the extreme case where the behavioral policy is uniform and therefore uninformative. The BPR efficiency therefore strongly relies on the assumption that the behavioral policy provides a beneficial inductive bias.

## 5.1 SAFE POLICY IMPROVEMENT

Conservative algorithms constrain the policy search to the set of policies $\Pi_{\text{PI}}$ for which the true policy improvement $\Delta_{\pi,\pi_{\hat{\beta}}} \doteq \mathcal{J}(\pi) - \mathcal{J}(\pi_{\hat{\beta}})$ can be safely estimated from $\hat{\Delta}_{\pi,\pi_{\hat{\beta}}} \doteq \hat{\mathcal{J}}(\pi) - \hat{\mathcal{J}}(\pi_{\hat{\beta}})$:

$$\forall \pi \in \Pi_{\text{PI}}, \quad \hat{\Delta}_{\pi,\pi_{\hat{\beta}}} - \Delta_{\pi,\pi_{\hat{\beta}}} \leq \epsilon_\Delta. \tag{5}$$

It is worth noting that the estimated behavior policy $\pi_{\hat{\beta}}$ necessarily belongs to $\Pi_{\text{PI}}$, since trivially $\hat{\Delta}_{\pi_{\hat{\beta}},\pi_{\hat{\beta}}} = \Delta_{\pi_{\hat{\beta}},\pi_{\hat{\beta}}} = 0$. Combined with BPR, conservative algorithms optimize on the policy set $\Pi_{\text{PI}} \cap \Pi_{\text{BPR}}$, and we consider the following relaxed assumptions, corresponding respectively to Assumptions 1.1 and 1.2:

**Assumption 2.** *(2.1) Access to an accurate estimate $\pi_{\hat{\beta}}$ of the true behavior $\pi_\beta$: $\mathcal{J}(\pi_\beta) - \mathcal{J}(\pi_{\hat{\beta}}) \leq \epsilon_\beta$. (2.2) Access to an accurate estimate $\hat{\Delta}_{\pi,\pi_{\hat{\beta}}}$ of the true policy improvement $\Delta_{\pi,\pi_{\hat{\beta}}}$ for all policies $\pi \in \Pi_{\text{PI}} \cap \Pi_{\text{BPR}}$: $\hat{\Delta}_{\pi,\pi_{\hat{\beta}}} - \Delta_{\pi,\pi_{\hat{\beta}}} \leq \epsilon_\Delta$.*

**Theorem 2.** *Under Assumption 2, BPR returns a policy $\pi_{\text{BPR}}$ with the following performance bounds: $\mathcal{J}_{\text{BPR}} \geq \mathcal{J}(\pi_\beta) + \hat{\Delta}_{\pi_{\text{BPR}},\pi_{\hat{\beta}}} - \epsilon_\Delta - \epsilon_\beta$.*

Assumption 2.1 amounts to behavior cloning (BC) guarantees. It is generally assumed as being an easier task than policy optimization, as BC is supervised learning. Theorem 4 provides bounds on $\epsilon_\beta$. Assumption 2.2 reflects the policy improvement objective. For instance, the SPIBB principle guarantees it with high probability in finite MDPs (Laroche et al., 2019b; Nadjahi et al., 2019; Simão et al., 2020). It is important to notice that Assumption 2.2 forbids, in theory, the use of the BPR representation to estimate the value directly. But in practice, we show that the representation learned by BPR is still useful to estimate the value function. We provide some experimental results in Appendix G.7 and a counterexample in Appendix F. Additionally, it is worth noting that ensuring that $\hat{\Delta}_{\pi_{\text{BPR}},\pi_{\hat{\beta}}}$ is positive is necessary to obtain approximate policy improvement guarantees. To do so, it suffices to fall back to $\pi_{\hat{\beta}}$ when the optimization of $\pi_{\text{BPR}}$ does not lead to a positive expected improvement $\hat{\Delta}_{\pi_{\text{BPR}},\pi_{\hat{\beta}}}$.

## 5.2 VALUE FUNCTION LOWER BOUND

In this section, we propose an alternative to Assumption 2.2 following the performance lower bound principle derived in pessimistic algorithms:

**Assumption 3.** *(3.1) Access to a lower bound of the performance $\mathcal{J}_\perp(\pi) \leq \mathcal{J}(\pi)$ for all policies, which is accurate for $\pi_{\hat{\beta}}$: $\mathcal{J}_\perp(\pi_{\hat{\beta}}) \geq \mathcal{J}(\pi_{\hat{\beta}}) - \epsilon_\perp$.*

In order to provide performance improvement bounds, we make use of the lower bound value gap $\Delta^\perp_{\pi,\pi_{\hat{\beta}}} \doteq \mathcal{J}_\perp(\pi) - \mathcal{J}_\perp(\pi_{\hat{\beta}})$.

**Theorem 3.** *Under Assumption 2.1 and 3.1, BPR returns a policy $\pi_{\text{BPR}}$ with the following performance bounds: $\mathcal{J}_{\text{BPR}} \geq \mathcal{J}(\pi_\beta) + \Delta^\perp_{\pi_{\text{BPR}},\pi_{\hat{\beta}}} - \epsilon_\perp - \epsilon_\beta$.*

Assumption 3.1 is satisfied by the Offline RL algorithm that is used in combination with BPR. For instance, CQL (Kumar et al., 2020; Yu et al., 2021) relies on the computation of a lower bound of the value function of the considered policies. More generally, pessimistic algorithms (Petrik et al., 2016; Yu et al., 2020; Kidambi et al., 2020; Jin et al., 2021; Buckman et al., 2021) have for principle to add an uncertainty-based penalty to the reward function, in order to control its risk to be overestimated. It is important to notice that, like for Assumption 2.2, this assumption would forbid (in theory) the use of the BPR representation to estimate the value.

Both approaches (safe policy improvement and value function lower bounds) suffer the same cost $\epsilon_\beta$, in comparison to their guarantees without BPR. The next subsection controls this quantity.

### 5.3    PERFORMANCE BOUND WITH BPR OBJECTIVE

In this section, we are concerned with the statistical property of the error between the estimated behavior based on the representation and the true behavior. To this end, we have the upper bound of the $\epsilon_\beta$ as:

**Theorem 4.** *With probability at least $1 - \delta$, for any $\delta \in (0, 1)$:*

$$\epsilon_\beta \le CK \cdot \frac{1}{n} \sum_{i=1}^{n} \left\| \pi_\beta(\cdot | s_i) - \pi_{\hat\beta}\left(\cdot \mid \phi(s_i)\right) \right\|_2 + 2\sqrt{2}K \cdot \mathrm{Rad}(\Phi) + K \cdot \sqrt{\frac{2\ln\frac{1}{\delta}}{n}} \qquad (6)$$

*where $n$ is the size of the dataset, $\mathrm{Rad}(\Phi)$ is the Rademacher complexity of $\phi$'s function class $\Phi$, $\pi_\beta$ is the behavior policy over the dataset, $C$ is a constant, and $K = \frac{R_{max}}{1-\gamma}$.*

Note that the first term in Equation 6 is the exact optimization problem of BPR in Equation 4 multiplied by constant $C \cdot K$, where the action $a_i$ is sampled from the offline dataset pairs $(s_i, a_i)$ whose behavior policy is $\pi_\beta$, and the predictor $f$ plays the similar role as $\pi_{\hat\beta}$. The second and the third term are both irrelevant to the specific representation, or the estimated behavior. This indicates that the potential harm of utilizing BPR can be reduced as the representation training procedure goes on.

## 6    EXPERIMENTS

BPR can be easily combined with *any existing* Offline RL pipeline. Our experimental results can be broken down into the following sections[3]:

- Does BPR outperform baseline Offline RL algorithms on standard raw-state inputs benchmarks?other representation objectives?

- Is BPR effective in learning policies on high-dimensional pixel-based inputs? Can BPR improve the robustness of the representation when the input contains complex distractors?

- Can BPR successfully improve the effective dimension of the feature in the value network?

### 6.1    EFFECTIVENESS OF BEHAVIOR REPRESENTATIONS IN D4RL OFFLINE BENCHMARK

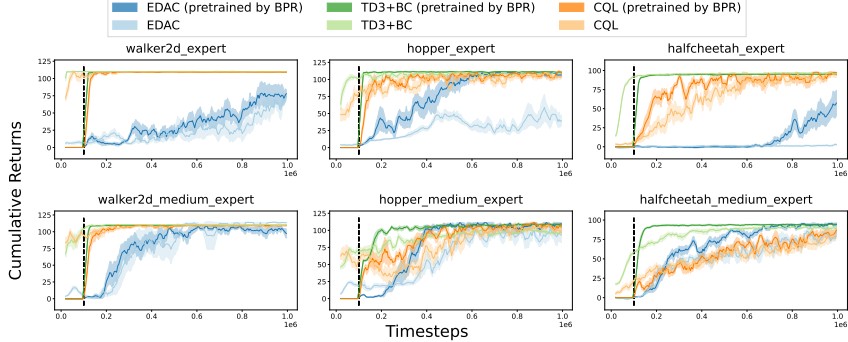

Figure 2: Performance comparison on D4RL dataset. The x-axis indicates the number of environment steps. The vertical axis reports the normalized cumulative returns. We train our 6 algorithms (*EDAC*, *TD3+BC*, *CQL*, and their BPR variants) on 6 seeds, and evaluate each one every 5,000 environment steps by computing the average return over 10 evaluation episodes. Lighter lines correspond to baselines, darker ones to versions pretrained with BPR. The black dotted line shows the time when representation pre-training ends and policy training process begins.

---

[3]Due to the page limit, we provide some additional experiment results in Appendix G.

**Performance Comparison in D4RL Benchmark** *Experimental Setup:* We analyze our proposed method BPR on the D4RL benchmark (Fu et al., 2020) of OpenAI gym MuJoCo tasks (Todorov et al., 2012) which includes a variety of datasets that have been commonly used in the Offline RL community. We evaluate our method by integrating it into three Offline RL methods: **TD3+BC** (Fujimoto & Gu, 2021), **CQL** (Kumar et al., 2020), and **EDAC** (An et al., 2021). We consider three environments: halfcheetah, hopper and walker2d, and two datasets per task: expert and medium-expert. Instead of passing the raw state as input to the value and policy networks as in the baseline methods, we first pretrain the encoder during $100k$ timesteps, then freeze it, pass the raw state through the frozen encoder to obtain its representation, and use that as the input of the Offline RL algorithm. Further details on the experiment setup are included in appendix G.

*Analysis:* Figure 2 shows the performance of all models on the D4RL tasks. We observe that pre-training the encoder with BPR leads to faster convergence for all Offline RL algorithms, and can improve policy performance. Notably, since different Offline RL algorithms share the same encoder network architecture, the pretrained BPR encoder can be reused across them, which substantially amortizes the pretraining cost.

**Performance Comparison among Different Representation Objectives** *Experimental Setup:* We follow the pre-training and fine-tuning paradigm of Yang & Nachum (2021), where we first pre-train the encoder for $100k$ timesteps, and then fine-tune a BRAC (Wu et al., 2019) agent on downstream tasks. We conduct performance comparison by training the encoder with different objectives, including BPR and several other state-of-the-art representation objectives[4].

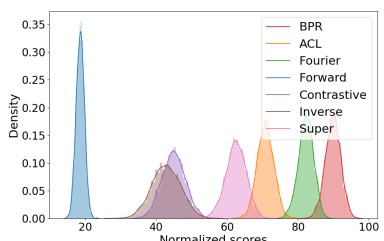

Figure 3: Bootstrapping distributions for uncertainty in IQM measurements with 5000 resamples

*Analysis:* We follow the performance criterion from Schwarzer et al. (2021b); Agarwal et al. (2021), and use the inter-quartile mean (IQM) normalized score, which is calculated overruns rather than tasks, and the percentile bootstrap confidence intervals, to compare the performance of the different objectives. The detailed results are provided in appendix G. Figure 3 presents the IQM normalized return and 95% bootstrap confidence intervals for all methods, the numerical values can be found in Table 1. The performance gain of BPR over the two most competitive baselines, Fourier and ACL, is statistically significant($p < 0.05$).

## 6.2 EFFECTIVENESS OF BEHAVIOR REPRESENTATIONS IN VISUAL OFFLINE RL BENCHMARK

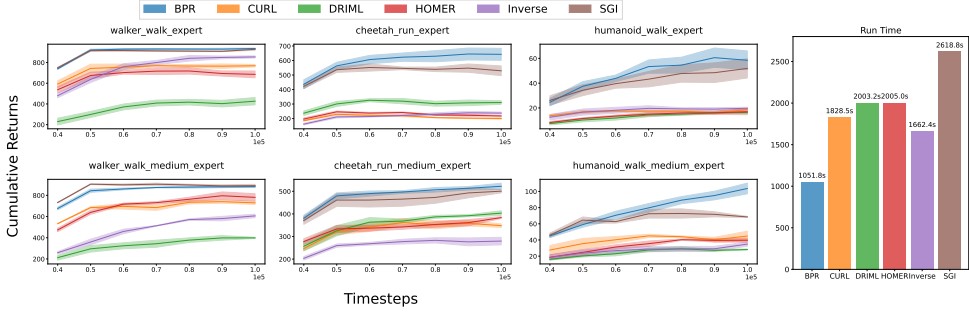

Figure 4: Performance comparison of BPR with several other baselines on DMC environments (left). Run time comparison for all representation objectives running on *cheetah_run_expert* task with 100K timesteps, the vertical axis indicates the wall-clock time in seconds (right).

---

[4]We use the codebase from Yang & Nachum (2021) in these experiments: `https://github.com/google-research/google-research/tree/master/rl_repr`

Table 1: Interquartile mean, median, and mean normalized scores, evaluated over all six runs for each of 16 DMC tasks. Confidence intervals computed by percentile bootstrap with 5000 resamples.

| Method | IQM | 95% CI | Median | 95% CI | Mean | 95% CI |
|---|---|---|---|---|---|---|
| BPR | **89.99** | (85.55, 93.97) | **87.04** | (77.04, 91.53) | **81.70** | (77.38, 86.21) |
| ACL | 70.79 | (66.09, 75.53) | 67.50 | (57.73, 77.28) | 69.91 | (65.98, 74.05) |
| Fourier | 82.24 | (77.86, 86.17) | 85.23 | (73.25, 88.99) | 72.61 | (68.01, 76.51) |
| Forward | 18.59 | (16.35, 20.90) | 9.26 | (6.68, 14.48) | 29.68 | (27.25, 32.20) |
| Contrastive | 44.87 | (38.36, 51.52) | 41.48 | (35.82, 54.39) | 43.56 | (39.18, 47.98) |
| Inverse | 42.79 | (34.97, 50.83) | 40.33 | (33.38, 47.47) | 44.58 | (39.20, 49.99) |
| Super | 62.62 | (56.81, 67.91) | 58.91 | (48.66, 68.79) | 60.72 | (57.08,64.53) |

**Performance comparison in V-D4RL benchmark**

**Experimental Setup**  We evaluate our method with five representation objectives that are considered as state-of-the-art methods on a benchmarking suite for Offline RL from visual observations of DMControl suite (DMC) tasks (Lu et al., 2022). We note that not all methods have been shown to be effective for visual-based continuous control problems. The baselines include: (i) Temporal contrastive learning methods such as DRIML (Mazoure et al., 2020) and HOMER (Misra et al., 2020); (ii) spatial contrastive approach, CURL (Laskin et al., 2020); (iii) one-step inverse action prediction model, Inverse model (Pathak et al., 2017); and (iv) Representation module which combines self-predictive, goal-conditioned RL and inverse model, namely SGI (Schwarzer et al., 2021a). We evaluate all representation objectives by integrating the pre-trained encoder from each into an Offline RL method DrQ+BC (Lu et al., 2022), which combines data augmentation techniques with the TD3+BC method (TD3 with a regularizing behavioral-cloning term to the policy loss).

**Analysis**  As shown in Figure 4, BPR consistently improves over the other state-of-the-art algorithms, except for SGI in the *walker_walk* task, while SGI is the most time-consuming representation objective of all methods. We evaluate the wall-clock training time of each representation objective for the 100K time steps. Our approach compares favorably against all previous methods. In particular, BPR is 2.5x faster than the most competitive SGI method. This suggests that the BPR objective can improve sample efficiency more effectively than the other representation objectives.

**Robustness to Visual Distractions**  *Experimental Setup:*  To test the policy robustness brought by BPR, we use three levels of distractors: easy, medium, and hard, to evaluate the performance of the model, following Lu et al. (2022). Each distraction represents a shift in the data distribution, where task-irrelevant visual factors are added (i.e., backgrounds, agent colors, and camera positions). Those factors of variation disturb the model training (see the demonstrations provided in Appendix G.8). We train the baseline agent, DrQ+BC, and its variant using a fixed encoder pretrained by BPR, on the three levels of the distraction of the *cheetah-run-medium-expert* dataset with 1 million data points. In the experiment, the agent is evaluated on three different test environments: i) *Original Env*, the evaluation environment is the original environment *i.e.*, without any distractors; ii) *Distractor Train*, the evaluation environment has the same distraction configuration as the training dataset, and the configuration is fixed during the evaluation procedure; iii) *Distractor Test*, the specific distractions of the evaluation environment changes over evaluation, while the level of the distractions remains the same.

Table 2: Cumulative return on *cheetah-run-medium-expert* dataset with visual distractions ranging from easy to hard. We compare DrQ+BC (before →) to DrQ+BC with an encoder pretrained with BPR (after →). The final performance is averaged over 6 seeds (highest result for each task underlined).

| Level of Distractions | Eval. Return Original | Eval. Return Dis. Train | Eval. Return Dis. Test |
|---|---|---|---|
| easy | 45.50($\pm$ 37.02) → 113.64($\pm$ 24.83) | 425.15($\pm$ 65.67) → 710.11($\pm$ 56.70) | 45.16($\pm$ 24.96) → 113.27($\pm$ 26.48) |
| medium | 37.35($\pm$ 25.66) → 104.30($\pm$ 22.95) | 586.17($\pm$ 99.66) → 702.00($\pm$ 90.44) | 14.70($\pm$ 5.93) → 27.53($\pm$ 11.62) |
| hard | 13.49($\pm$ 10.66) → 64.79($\pm$ 33.47) | 497.41($\pm$ 103.85) → 608.98($\pm$ 64.64) | 18.88($\pm$ 11.32) → 16.72($\pm$ 6.88) |

*Analysis:*  The final performances are shown in Table 2. As can be seen, BPR improves the policy performance in most cases, except for the task with a combination of the *hard level* and the *distractor*

*test* setting where the averaged performance of the agent with BPR is close to the one without BPR but has a lower variance. Besides, the different evaluation environment corresponds to different degrees of the distribution shift from the training dataset. It is therefore not surprising to see that the agent performs better in *Distractor train* environment than the others. This experiment shows that much better (more robust) policies can be obtained against distractions with the help of BPR, indicating that BPR objective can somehow disentangle sources of distractions through training, facilitating the learning of general features.

### 6.3 EFFECTIVE DIMENSION OF THE VALUE NETWORK

***Experimental Setup:*** We investigate whether utilizing the fixed encoder learned from BPR can alleviate the "under-parameterization" phenomenon of the value network. To this end, we propose to use the effective dimension (defined in Definition 1) as a way to evaluate the compactness of features represented in the value network. We first sample a batch of state-action pairs from the offline dataset and then compute the effective dimension of the output of the penultimate layer of the value network. In this experiment, we evaluate the effective dimension of the value network of the CQL agent on *Halfcheetah-medium-expert-v2* task in D4RL datasets, which operates on the physical meaning raw state inputs. For comparison, we also develop a variant of CQL whose input is still the raw state, but a shared encoder head is applied to both the value network and policy network and is trained with the critic loss along the policy learning procedure.

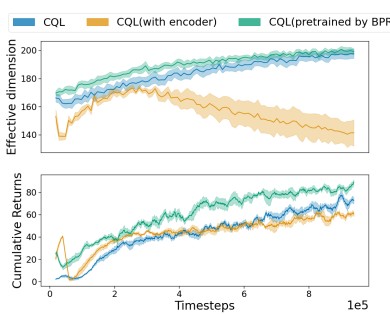

Figure 5: Effective dimension of the penultimate layer of the value network (top) and performance (bottom) over the training on the Halfcheetah-medium-expert-v2 task.

***Analysis:*** Figure 5 illustrates that over the course of training, BPR objective successfully improves the effective dimensions of the state representation, meanwhile achieving better performance compared to the agent that is without an encoder and the agent whose encoder is trained with the critic loss. Notably, when training the encoder via the critic loss along the policy learning procedure, the effective dimension increases significantly early in training, suggesting that even without any specific representation loss, the encoder can still disentangle some similar states. When the training goes on, the agent will observe more diverse states over time as the policy improves, which induces a mismatch between the learnt representations and the newly observed states, leading to a decline in the effective dimension. In contrast, with a fixed encoder learned using the BPR objective, CQL produces more effective dimensions than its vanilla version, leading to better performance, which indicates that BPR is capable of providing compact information for downstream Offline RL training.

## 7 DISCUSSION

**Limitations and Future Work**  Although this work has shown effectiveness in learning representations that are robust to visual distractions, it still struggles when the distribution of evaluation environments shifts too much from the training environment, this suggests that further improvement on the generalization abilities of the representation is required. Another limitation related to the nature of the approach is that if the behavioral policy is uninformative, then our approach will likely result in a policy that has the same performance as the behavior but without any improvement. However, it is straightforward to leverage information about the behavior (either a priori or from statistics on the returns in the dataset) to decide when to use BPR. From these perspectives, further development of a more theoretically grounded representation objective might be needed.

**Conclusion**  Offline RL algorithms typically suffer from two main issues: one is the difficulty of learning from high-dimensional visual input data, the other is the implicit under-parameterization phenomenon that can lead to low-rank features of the value network, further resulting in low performance. In this work, we propose a simple yet effective method for state representation learning in an Offline RL setting which can be combined with any existing Offline RL pipeline with minimal changes.

## 8 REPRODUCIBILITY STATEMENT

To ensure the reproducibility of all empirical results, we provide the code base in the supplementary material, which contains: training scripts, and requirements for the virtual environment. All datasets and code platforms (PyTorch and Tensorflow) we use are public. To rebuild the architecture of BPR model and plug BPR in any Offline RL algorithms, the readers are suggested to check the descriptions in Section 4, especially the **Implementation details** paragraph. All proofs are stated in Appendix E with explanations and underlying assumptions. We also provide the pseudocode of the pretraining process and the co-training process in Algorithm 1 and 2 in Appendix.

All training details are specified in Section 6 and Appendix G. In the experiment on d4rl tasks, all representation objectives use the same encoder architecture, i.e., with 4-layer MLP activated by ReLU, followed by another linear layer activated by Tanh, where the final output feature dimension of the encoder is 256. Besides, all representation objectives follow the same optimizer settings, pre-training data, and the number of pre-training epochs. In the experiment on v-d4rl tasks, all representation objectives use the same encoder architecture, i.e., with four convolutional layers activated by ReLU, followed by a linear layer normalized by LayerNorm and activated by Tanh, where the final output feature dimension of the encoder is 256. We also provide the pseudocode for each visual representation objective in Algorithm 4-7 in Appendix.

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

# Appendix

## CONTENTS

## A   NOTATION

Table 3 summarizes our notation.

Table 3: Table of Notation.

| Notation | Meaning | Notation | Meaning |
|----------|---------|----------|---------|
| $\phi$ | state encoder network | $\mathcal{A}$ | action space |
| $f$ | predictor network | $\mathcal{Z}$ | state representation space |
| $\rho$ | initial state distribution | $n_s$ | dimension of state space |
| $\zeta$ | effective dimension | $n_a$ | dimension of action space |
| $\gamma$ | discounted factor | $T$ | transition probability function |
| $s$ | state | $\mathcal{D}$ | offline dataset |
| $a$ | action | $\Psi$ | feature matrix |
| $r$ | reward | $\pi_\beta(a\|s)$ | behavior policy of offline dataset |
| $d$ | the feature dimension | $\pi_{\hat{\beta}}(a\|s)$ | the estimated behavior policy |
| $y$ | state prediction | $d^{\pi_\beta}(s)$ | discounted state occupancy density |
| $z$ | state representation | $V^\pi(s)$ | state value function |
| $n$ | the numbers of datapoints | $\mathcal{J}(\pi)$ | the performance of policy $\pi$ |
| $\mathcal{S}$ | state space | $\mathcal{J}_\perp(\pi)$ | the lower bound of the performance |

## B   ADDITIONAL RELATED WORK

**Representation learning in Online RL**    State representation learning lies at the heart of the empirical successes of deep RL. Traditionally, state representation learning has been framed as learning state abstractions / aggregations (Andre & Russell, 2002; Ferns et al., 2006; Mannor et al., 2004; Comanici et al., 2012), and most of these methods aim to reduce the original state space size and to minimize the system complexity. Recent studies present promising results in learning robust representations that can then be used to accelerate policy learning in pixel-based observation spaces. Lee et al. (2020); Yarats et al. (2021b) propose to train deep auto-encoders with a reconstruction loss to learn compact representations, Shelhamer et al. (2017); Hafner et al. (2019) learn state representations from predictive losses to maintain trajectory consistency, van den Oord et al. (2018); Laskin et al. (2020); Stooke et al. (2021) use contrastive losses as auxiliary tasks to improve performance, Xu et al. (2014); Krishnamurthy et al. (2016); Yarats et al. (2021a) develop state clustering methods to improve sample-efficiency, and finally Zhang et al. (2021a); Castro et al. (2021); Zang et al. (2022) measure state distance by bisimulation similarity to shape state representations.

## C   COMPARISON

### C.1   BEHAVIOR CLONING

One would consider BPR similar to the behavior cloning method that imitates the behavior policy of the data since they both learn a projection from state to action. Despite resembling behavior cloning in form, we emphasize that the goal of BPR is not to approximate the exact behavior policy. Instead, it is a two-stage process that reutilizes the learned encoder on downstream tasks to accelerate policy learning, while discarding the initial projection layer. Additionally, this allows us to apply $\ell_2$-normalization to the prediction and action rather than using their true values. This normalization has been shown to be important for representation learning (Wang & Isola, 2020; Grill et al., 2020; Zang et al., 2022). The difference is illustrated in Figure 6.

### C.2   $\pi^*$-IRRELEVANCE ABSTRACTION

A $\pi^*$-irrelevance abstraction $\phi_{\pi^*}$ (Jong & Stone, 2005; Li et al., 2006) is such that every abstract class has an action $a^*$ that is optimal for all the states in that class that is $\phi_{\pi^*}(s_1) = \phi_{\pi^*}(s_2)$ implies that $Q^*(s_1, a^*) = \max_a Q^*(s_1, a)$ and $Q^*(s_2, a^*) = \max_a Q^*(s_2, a)$, which attempt to preserve

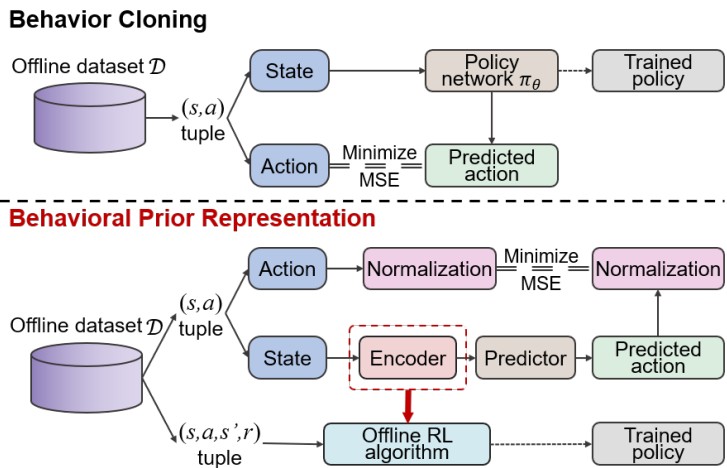

Figure 6: A summary of the BPR algorithm shows its difference from Behavior Cloning. BPR focus on learning state representation to benefit the downstream Offline RL algorithms instead of concentrating on learning the behavior policy.

the optimal action[5]. In Li et al. (2006), $\pi^*$-irrelevance abstraction was applied on Q-learning and was proven that the induced value function could not converge to the optimal in the ground MDP. While comparing with BPR, the biggest difference is that: in theory, as described in Assumption 2.2, we use the BPR representation only for the policy search, while Li et al. (2006) assumes that the value function should be estimated for learning reasonable Q value in representation space. As a consequence, Li et al. (2006) requires much stronger assumptions for the policy abstractions, such as bisimulation or invariance for the optimal value (policy). As an example, consider an MDP as shown in Figure 7 (an example is taken from the MDP investigated by Li et al. (2006)): for $\pi^*$-irrelevance abstraction, it induces an abstract MDP $\bar{M} = < \bar{S}, A, \bar{P}, \bar{R}, \gamma >$, where we apply the q learning on the induced $\bar{M}$. Under this abstract MDP, since $S_1$ and $S_2$ are accidentally aggregated to one state abstraction, the abstract reward $\bar{R}$ and the abstract transition $\bar{P}$ will be estimated mistakenly, therefore the q value cannot be updated to the optimal. Notably, the value estimation we perform on the original input space, which belongs to the ground MDP state space, differs from the value estimation on the representation space that raises issues that can only be alleviated with the drastic assumptions in Li et al. (2006).

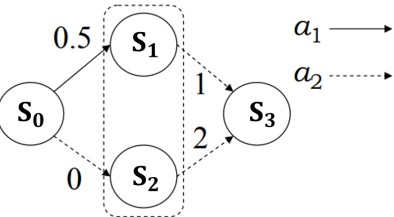

Figure 7: The solid and dashed lines represent actions 1 and 2, respectively. The corresponding graph shows the value function in the aggregated (middle) state for each of its actions. $\phi_{\pi^*}$ yields an optimal policy for $\bar{M}$ that is suboptimal in $M$, while BPR can still find the optimal policy in $M$ by policy search.

---

[5]To keep the notation aligned well with Li et al. (2006), we will replace $\pi^\star$ with $\pi^*$, and replace $\mathcal{M}$ with $M$ in this section.

### C.3 FOURIER

Nachum & Yang (2021a) developed a representation objective that combines contrastive learning with the linear dynamic model (i.e., a learnable function whose input is the pair of next state and action $(s', a)$, the output is a representation of the current state), where the contrastive objective is approximated by leveraging the Random Fourier Features. Though effective, this method assumes the underlying dynamic model is linear, while the BPR objective does not show reliance on such an assumption. Since the BPR objective can be oblivious to the reward function and transition model, it can be applied to a wild range of domains where the reward or the transition could be hard to approximate. On the other hand, without modeling the reward function or transition dynamics, the BPR objective is slightly theoretically weaker (see the counterexample in the Appendix). A combination of BPR and the methods like Fourier should be further investigated in future work, which will provide a more theoretically grounded yet still simple approach.

### C.4 ACL

Yang & Nachum (2021) introduced Attentive Contrastive Learning (ACL) to learn state representation, which uses the transformer-based architecture as a skeleton, where a subset of sub-trajectories are randomly masked to make predictions. Similar to the Fourier approach described above, ACL also utilizes a contrastive learning objective to update its feature mapping, while further reconstructing the action and the reward to stabilize the training. BPR, on the other hand, can be seen as a representation objective that, with only a reconstruction module for predicting actions, is far more simple than the transformer-based module that requires sequential prediction. Another potential benefit of the BPR objective is that it can possibly integrate into other kinds of representation objectives due to its independence from the encoder architecture.

## D EFFECTIVE DIMENSION AND FEATURE RANK

The desirable state representation should not only be able to guide a good policy learning procedure, but also be compact enough to provide concise yet effective information. We focus on measuring the "compactness" of the state representation to show that BPR suffices for the agent to benefit from learning representation with auxiliary representation loss, which can alleviate the "implicit under-parameterization" phenomenon.

**Assumption 4.** *(Feature reachability) Denote $\lambda_{\min}(A)$ as the smallest eigenvalue of a positive-definite matrix A. With a mapping function $\phi^\star$, we assume that there exists $\epsilon \in \mathbb{R}^+$ such that,*

$$\sup_\pi \lambda_{\min} \left( \mathbb{E}_{s \sim d^{\pi_\beta}(s)} \left[ \phi^\star(s)\phi^\star(s)^\top \right] \right) \geq \epsilon. \tag{7}$$

Assumption 4 posits that in MDP $\mathcal{M}$, for each latent state in representation space $\mathbb{R}^d$, there exists a policy that reaches it with a non-zero probability. This ensures the optimal state representation should have no redundant dimensions, which is a reasonable assumption for a compact latent space. While a similar assumption is commonly made in previous work (Agarwal et al., 2022; Modi et al., 2021), seldom of them consider leveraging it to measure the feature effectiveness. Based on this assumption, we define the following measurement, inspired by the Feature rank defined in Lyle et al. (2022):

**Definition 2.** *Effective Dimension Let $\mathrm{ED}(M)$ denote the multiset of eigenvalues of a square matrix M. Then the effective dimension of $\Psi$ is defined to be*

$$\zeta(\Psi, \mathcal{D}, \epsilon) = \mathbb{E}_{\mathcal{D}} \left[ \left| \left\{ \sigma \in ED \left( \left( \frac{1}{\sqrt{|\mathcal{S}||\mathcal{A}|}} \Psi^\top \right) \left( \frac{1}{\sqrt{|\mathcal{S}||\mathcal{A}|}} \Psi \right) \right) | \sigma > \epsilon \right\} \right| \right], \tag{8}$$

*where $\Psi \in \mathbb{R}^{|\mathcal{S}||\mathcal{A}| \times d}$ is the feature matrix with respect to the output of the penultimate layer of the state-action value network.*

**Lemma 5.** *Let $\mathbf{X}_n \subset \mathbb{R}^{|\mathcal{S}||\mathcal{A}|}$ be a set of $n$ state-action pairs in $\mathbb{R}^{|\mathcal{S}||\mathcal{A}|}$ sampled from a fixed distribution $d^{\pi_\beta}(s, a)$, and the corresponding feature matrix being $\Psi_n$, a consistent estimator can be conducted as:*

$$\hat{\zeta}(\Psi, \mathcal{D}, \epsilon) = \left| \left\{ \sigma \in ED \left( \left( \frac{1}{\sqrt{n}} \Psi_n \right)^\top \left( \frac{1}{\sqrt{n}} \Psi_n \right) \right) | \sigma > \epsilon \right\} \right|. \tag{9}$$

*Proof.* The following proof mimics the derivation in Lyle et al. (2022). Recall that

$$\left(\frac{1}{\sqrt{n}}\Psi_n\right)^\top \left(\frac{1}{\sqrt{n}}\Psi_n\right) = \frac{1}{n}\sum_{i=1}^n \psi(x_i)\psi(x_i)^\top. \tag{10}$$

The following property of the expected value holds

$$\mathbb{E}_{\mathcal{D}}\left[\psi(x)\psi(x)^\top\right] = \mathbb{E}\left[\frac{1}{n}\sum_{i=1}^n \psi(x_i)\psi(x_i)^\top\right]. \tag{11}$$

Then, consider each element of the matrix $M$:

$$\mathbb{E}\left[\left(\psi(x)\psi(x)^\top\right)_{ij}\right] = M_{ij} = \mathbb{E}\left[\psi_i(x)\psi_j(x)\right] \Longrightarrow \sum_{k=1}^n \frac{1}{n}\psi_i(x_k)\psi_j(x_k) \overset{a.s.}{\to} M_{ij} \tag{12}$$

Since we have convergence for any $M_{ij}$, we get convergence of the resulting matrix to $M$. And since the eigenvalues are continuous functions of $M$, the eigenvalues of $M_n$ converge to those of $M$ almost surely. $\square$

Intuitively, it possesses a similar equivalent form of feature reachability. Empirically, we set $\epsilon$ as 0.01 where only higher eigenvalues are considered, eliminating the distraction caused by small noise. With the effective dimensions measurement, we can show that BPR can stabilize the training procedure, and provides a higher effective feature dimension for the value network, thus helping to alleviate the "implicit under-parameterization" phenomenon. Notably, Equation 8 can also be used to measure the effective dimension of the state representation, i.e., $\zeta(\Phi, \mathcal{D}, \epsilon)$, when we substitute the feature matrix $\Psi$ by the representation matrix $\Phi \in \mathbb{R}^{|\mathcal{S}|\times|\mathcal{Z}|}$ where $\mathcal{Z}$ is the latent state representation space.

**Connection to the effective dimension proposed in Lan et al. (2022)** To develop a measurement of the effective dimension of state representation, Lan et al. (2022) borrow the concept of $\mu$-coherence (Candès & Recht, 2009; Mohri & Talwalkar, 2011), which is related to the statistical leverage scores (Drineas et al., 2006; Mahoney et al., 2012).

**Definition 3.** *Given an arbitrary $n \times d$ matrix $A$, with $n > d$ and $rank(A) = r$, let $U$ denote the $n \times d$ matrix consisting of the $d$ left singular vectors of $A$, and let $U_{(i)}$ denote the $i$-th row of the matrix $U$ as a row vector. Then the statistical leverage scores are given by $\|U_{(i)}\|_2^2$, for $i \in \{1, \cdots, n\}$; the coherence $\mu$ is:*

$$\mu(U) = \max_{i\in\{1,\cdots,n\}} \|U_{(i)}\|^2, \tag{13}$$

*i.e., the largest statistical leverage score. When we let $P_U$ be the orthogonal projection onto $U$ and $(e_i)$ the canonical basis, we can define $\mu'$-coherence as:*

$$\begin{aligned} \mu'(U) &= n \max_{i\in\{1,\cdots,n\}} \|P_U e_i\|^2 = n \max_{i\in\{1,\cdots,n\}} \|UU^T e_i\|^2 \\ &= n \max_{i\in\{1,\cdots,n\}} \|U^T e_i\|^2 = n \max_{i\in\{1,\cdots,n\}} \|U_{(i)}\|^2. \end{aligned} \tag{14}$$

Following the above definition, Lan et al. (2022) defined the effective dimension as:

**Definition 4** (Effective dimension in Lan et al. (2022)). *Let $\Phi \in \mathbb{R}^{S \times k}$ be a feature matrix. The effective dimension of $\Phi$ (vis-a-vis the standard basis $(e_i)$) is defined as the quantity*

$$d_{\text{eff}}(\Phi) := S \max_{i=1,\dots,S} \|P_\Phi e_i\|_2^2, \tag{15}$$

*where $P_\Phi$ is the orthogonal projector onto the column space of $\Phi$, $S$ is the number of the state.*

Note that for any feature matrix, the smallest $d_{\text{eff}}$ can be the rank of the state space, achieved, for example, if $\Phi$ is spanned by vectors whose entries all have magnitude $1/\sqrt{S}$, meaning that the feature matrix is perfectly learned. The largest possible value for $d_{\text{eff}}$ is $S$, which would correspond to any subspace that contains a standard basis element, meaning that the state space is full-rank.

As a comparison, effective dimension in this paper and the one in Lan et al. (2022) are both developed from the Singular Value Decomposition (SVD) that was generally utilized for dimensionality reduction[6], the difference is that we compute the number of the eigenvalue of the Gram matrix of the feature matrix as the measurement, while Lan et al. (2022) compute the largest leverage score of the feature matrix that considered as the left singular vectors of the state space. In some sense, the effective dimension in Lan et al. (2022) can be seen as a variation of Equation 8 with $\epsilon = 0$, which also indicates that the small disturbance component of the feature matrix cannot be discarded in Equation 15 without prior knowledge of the rank of the state space.

## E  PROOFS

### E.1  USEFUL TECHNICAL RESULTS

We begin this section by introducing Rademacher complexity, which can be used to derive data-dependent upper bounds on the learnability of function classes.

**Definition 5.** *Let $\mathcal{X}$ be any set, $X_1, X_2, \cdots, X_n$ be i.i.d. random variables with values in $\mathcal{X}$. We have $\mathcal{G}$ a class of functions $g : \mathcal{X} \to \mathbb{R}$. The quality*

$$\mathrm{Rad}(\mathcal{G}) := \mathbb{E}_{X_j,\sigma_j}\left[\sup_{g\in\mathcal{G}}\frac{1}{n}\sum_{j=1}^{n}\sigma_j g\left(X_j\right)\right] \tag{16}$$

*is called Rademacher complexity of the class $\mathcal{G}$ on the sample $X = (X_1, X_2, \cdots, X_n) \in \mathcal{X}^n$, where $\sigma$ is the Rademacher random variables that are drawn uniformly at random from $\{\pm 1\}$.*

The corresponding vectorized version of Rademacher complexity is:

$$\mathrm{Rad}(\mathcal{G}) := \mathbb{E}_{X_j,\sigma_{ji}}\left[\sup_{g\in\mathcal{G}}\frac{1}{n}\sum_{j=1}^{n}\sum_{i=1}^{K}\sigma_{ji} g\left(X_j\right)_i\right], \tag{17}$$

where $\mathcal{G}$ is a class of functions $g : \mathcal{X} \to \mathbb{R}^K$.

### E.2  THEORETICAL ANALYSIS

**Assumption 1.** *(1.1) Access to the true behavior: $\pi_{\hat{\beta}} = \pi_\beta$. (1.2) Access to the true performance of policies: $\hat{\mathcal{J}}(\pi) = \mathcal{J}(\pi)$ for all policies $\pi$. (1.3) The embedding $\phi$ allows to represent the behavior policy estimate: $\pi_{\hat{\beta}}(a|\phi(s)) = \pi_{\hat{\beta}}(a|s) \in \Pi_{\mathrm{BPR}}$. (1.4) The algorithm performs perfect optimization: $\mathcal{J}_{\mathrm{BPR}} = \max_{\pi\in\Pi_{\mathrm{BPR}}} \hat{\mathcal{J}}(\pi)$.*

**Theorem 2.** *Under idealized Assumption 1.1-1.4, BPR returns a policy that improves over the behavior policy: $\mathcal{J}_{\mathrm{BPR}} \geq \mathcal{J}(\pi_\beta)$.*

*Proof.*

$$\mathcal{J}_{\mathrm{BPR}} = \mathcal{J}\left(\arg\max_{\pi\in\Pi_{\mathrm{BPR}}}\hat{\mathcal{J}}(\pi)\right) \qquad\qquad \text{(Assumption 1.4)}$$

$$= \max_{\pi\in\Pi_{\mathrm{BPR}}}\mathcal{J}(\pi) \qquad\qquad\qquad\quad \text{(Assumption 1.2)}$$

$$\geq \mathcal{J}(\pi_{\hat{\beta}}) \qquad\qquad\qquad\qquad \text{(Assumption 1.3)}$$

$$\geq \mathcal{J}(\pi_\beta) \qquad\qquad\qquad\qquad \text{(Assumption 1.1)}$$

which concludes the proof. $\qquad\qquad\qquad\qquad\qquad\qquad\qquad\qquad\qquad\qquad\qquad\qquad \square$

**Assumption 2.** *(2.1) Access to an accurate estimate $\pi_{\hat{\beta}}$ of the true behavior $\beta$: $\mathcal{J}(\pi_\beta) - \mathcal{J}(\pi_{\hat{\beta}}) \leq \epsilon_\beta$. (2.2) Access to an accurate estimate $\hat{\Delta}_{\pi,\pi_{\hat{\beta}}}$ of the true policy improvement $\Delta_{\pi,\pi_{\hat{\beta}}}$ for all policies $\pi \in \Pi_{\mathrm{PI}} \cap \Pi_{\mathrm{BPR}}$: $\hat{\Delta}_{\pi,\pi_{\hat{\beta}}} - \Delta_{\pi,\pi_{\hat{\beta}}} \leq \epsilon_\Delta$.*

---

[6]the singular values of the feature matrix are the eigenvalues of $M$ in Equation 12 when we only consider the state feature

**Theorem 3.** *Under Assumption 2, BPR returns a policy $\pi_{\mathrm{BPR}}$ with the following performance bounds: $\mathcal{J}_{\mathrm{BPR}} \geq \mathcal{J}(\pi_\beta) + \hat{\Delta}_{\pi_{\mathrm{BPR}}, \pi_{\hat{\beta}}} - \epsilon_\Delta - \epsilon_\beta$.*

*Proof.*

$$
\begin{aligned}
\mathcal{J}_{\mathrm{BPR}} = \mathcal{J}(\pi_{\mathrm{BPR}}) && \text{(by definition)} \\
\geq \mathcal{J}(\pi_{\hat{\beta}}) + \hat{\Delta}_{\pi_{\mathrm{BPR}}, \pi_{\hat{\beta}}} - \epsilon_\Delta && \text{(Assumption 2.2)} \\
\geq \mathcal{J}(\pi_\beta) + \hat{\Delta}_{\pi_{\mathrm{BPR}}, \pi_{\hat{\beta}}} - \epsilon_\Delta - \epsilon_\beta && \text{(Assumption 2.1)}
\end{aligned}
$$

which concludes the proof. $\qquad\square$

**Assumption 3.** *(3.1) access to a lower bound of the performance $\mathcal{J}_\perp(\pi) \leq \mathcal{J}(\pi)$ for all policies, which is tight for $\hat{\beta}$: $\mathcal{J}_\perp(\pi_{\hat{\beta}}) \geq \mathcal{J}(\pi_{\hat{\beta}}) - \epsilon_\perp$.*

**Theorem 4.** *Under Assumptions 2.1 and 3.1, BPR returns a policy $\pi_{\mathrm{BPR}}$ with the following performance bounds: $\mathcal{J}_{\mathrm{BPR}} \geq \mathcal{J}(\pi_\beta) + \Delta^\perp_{\pi_{\mathrm{BPR}}, \pi_{\hat{\beta}}} - \epsilon_\perp - \epsilon_\beta$.*

*Proof.*

$$
\begin{aligned}
\mathcal{J}_{\mathrm{BPR}} = \mathcal{J}(\pi_{\mathrm{BPR}}) && \text{(by definition)} \\
\geq \mathcal{J}_\perp(\pi_{\mathrm{BPR}}) && \text{(Assumption 3.1)} \\
= \mathcal{J}_\perp(\pi_{\hat{\beta}}) + \Delta^\perp_{\pi_{\mathrm{BPR}}, \pi_{\hat{\beta}}} && \text{(by definition)} \\
\geq \mathcal{J}(\pi_{\hat{\beta}}) + \Delta^\perp_{\pi_{\mathrm{BPR}}, \pi_{\hat{\beta}}} - \epsilon_\perp && \text{(Assumption 3.1)} \\
\geq \mathcal{J}(\pi_\beta) + \Delta^\perp_{\pi_{\mathrm{BPR}}, \pi_{\hat{\beta}}} - \epsilon_\perp - \epsilon_\beta && \text{(Assumption 2.1)}
\end{aligned}
$$

which concludes the proof. $\qquad\square$

### E.3 PERFORMANCE BOUND WITH BPR OBJECTIVE

BPR solve the optimization problem:

$$
\min \frac{1}{n} \sum_{i=1}^{n} \|f(\phi(s_i)) - a_i\|_2 : \phi \in \Phi, f \in \mathcal{F}, \tag{18}
$$

where $(s_i, a_i)$ is an i.i.d. sample from the offline dataset with the size of $n$.

The difference between the estimated behavior policy and the true behavior policy is:

$$
\mathbb{E}\left[J(\pi_{\hat{\beta}}) - J(\pi_\beta) | \mathcal{D}\right] = \mathbb{E}_{s \sim \rho}\left[V^{\pi_{\hat{\beta}}}(s) - V^{\pi_\beta}(s)\right]. \tag{19}
$$

**Lemma 5.** *Consider a fixed dataset $\mathcal{D}$ and the reward is bounded by $R_{max}$, then the difference between $\pi_{\hat{\beta}}$ and $\pi_\beta$ will be bounded as:*

$$
\mathbb{E}\left[J(\pi_{\hat{\beta}}) - J(\pi_\beta) | \mathcal{D}\right] \leq \frac{2R_{max}}{(1-\gamma)^2} \cdot \mathbb{E}\left[D_{\mathrm{TV}}\left(\pi_{\hat{\beta}}(s)\|\pi_\beta(s)\right) | \mathcal{D}\right] \tag{20}
$$

*Proof.* Recall that the total variation distance (TVD) between the estimated policy $\pi_{\hat{\beta}}$ and the behavior policy $\pi_\beta$ for a given state $s$ is defined as:

$$
D_{\mathrm{TV}}(\pi_{\hat{\beta}}(s), \pi_\beta(s)) = \sup_{a \in \mathcal{A}} \left|\pi_{\hat{\beta}}(a|s) - \pi_\beta(a|s)\right| = \frac{1}{2} \sum_a \|\pi_{\hat{\beta}}(a|s) - \pi_\beta(a|s)\|_1. \tag{21}
$$

We expand the expectation in the value bound:

$$
\mathbb{E}_{s \sim \rho}\left[V^{\pi_{\hat{\beta}}}(s) - V^{\pi_\beta}(s)\right]
$$

$$
\leq \sum_s \rho(s) \left(\sum_a \left|\pi_{\hat{\beta}}(a \mid s) - \pi_\beta(a \mid s)\right| \left(\mathcal{R}(s,a) + \gamma \sum_{s'} T(s,a,s') |V^{\pi_{\hat{\beta}}}(s') - V^{\pi_\beta}(s')|\right)\right)
$$

$$
= \sum_s \sum_a \rho(s) \left|\pi_{\hat{\beta}}(a \mid s) - \pi_\beta(a \mid s)\right| \left(\mathcal{R}(s,a) + \gamma \sum_{s'} T(s,a,s') |V^{\pi_{\hat{\beta}}}(s') - V^{\pi_\beta}(s')|\right).
$$

$$
\tag{22}
$$

Apply the upper bound of the value function $R_{\max}/(1-\gamma)$ to the above equation:

$$
\begin{aligned}
\mathbb{E}\left[J(\pi_{\hat{\beta}}) - J(\pi_{\beta})|\mathcal{D}\right] &= \mathbb{E}_{s\sim\rho}\left[V^{\pi_{\hat{\beta}}}(s) - V^{\pi_{\beta}}(s)\right] \\
&\leq \frac{R_{\max}}{1-\gamma}\mathbb{E}_{s\sim\rho}\left[\sum_a \left|\pi_{\hat{\beta}}(a\mid s) - \pi_{\beta}(a\mid s)\right|\right] \\
&= \frac{2R_{\max}}{1-\gamma}\cdot\mathbb{E}\left[\mathbb{E}_{s\sim\rho}\left[D_{\mathrm{TV}}\left(\pi_{\hat{\beta}}(s)\|\pi_{\beta}(s)\right)\right]|\mathcal{D}\right] \\
&= \frac{2R_{\max}}{(1-\gamma)^2}\mathbb{E}_{s\sim d^{\pi_\rho}}\left[D_{\mathrm{TV}}\left(\pi_{\hat{\beta}}(s)\|\pi_{\beta}(s)\right)\right].
\end{aligned}
\tag{23}
$$

For simplicity, we denote $\mathbb{E}_{s\sim d^{\pi_\rho}}\left[D_{\mathrm{TV}}\left(\pi_{\hat{\beta}}(s)\|\pi_{\beta}(s)\right)\right]$ as $\mathbb{E}\left[D_{\mathrm{TV}}\left(\pi_{\hat{\beta}}(s)\|\pi_{\beta}(s)\right)|\mathcal{D}\right]$. $\qquad\square$

This lemma tells us the suboptimality of an arbitrary policy is upper-bounded by the TVD between the optimal policy and itself over the dataset $\mathcal{D}$.

Consider we learn the representation mapping $\phi : \mathcal{S} \to \mathcal{Z}$, and we learn a policy $\pi_\phi : \mathcal{Z} \to \mathcal{A}$ based on the fixed mapping $\phi$, instead of learning the policy $\pi : \mathcal{S} \to \mathcal{A}$. The following theorem will provide an upper bound of the suboptimality of the policy $\pi_\phi$[7].

**Theorem 5.** *With probability at least $1-\delta$, for any $\delta \in (0,1)$:*

$$
\epsilon_\beta = \mathbb{E}\left[J(\pi_{\hat{\beta}}) - J(\pi_{\beta})\,\middle|\,\mathcal{D}\right] \leq CK\cdot\frac{1}{n}\sum_{i=1}^n \left\|\pi_\beta(\cdot|s_i) - \pi_{\hat{\beta}}\left(\cdot\mid\phi(s_i)\right)\right\|_2
$$
$$
+ 2\sqrt{2}K\cdot\mathrm{Rad}(\Phi) + K\cdot\sqrt{\frac{2\ln\frac{1}{\delta}}{n}}
\tag{24}
$$

*where $n$ is the size of the dataset, $Rad$ is the Rademacher complexity, $\pi_\beta$ is the behavior policy over the dataset, $C$ is a constant, $K = \frac{R_{max}}{1-\gamma}$.*

*Proof.* The following proves are build on techniques provided by Asadi et al. (2020). First, for $\forall\phi$, we have:

$$
\mathbb{E}_{s\sim d^{\pi_\beta}}\left[\left\|\pi_\beta(\cdot\mid s) - \pi_{\hat{\beta}}(\cdot\mid\phi(s))\right\|_1\right] - \frac{1}{n}\sum_{i=1}^n\left\|\pi_\beta(\cdot\mid s_i) - \pi_{\hat{\beta}}(\cdot\mid\phi(s_i))\right\|_1 \leq
$$
$$
\underbrace{\sup_{\phi\in\Phi}\left\{\mathbb{E}_{s\sim d^{\pi_\beta}}\left[\left\|\pi_\beta(\cdot\mid s) - \pi_{\hat{\beta}}(\cdot\mid\phi(s))\right\|_1\right] - \frac{1}{n}\sum_{i=1}^n\left\|\pi_\beta(\cdot\mid s_i) - \pi_{\hat{\beta}}(\cdot\mid\phi(s_i))\right\|_1\right\}}_{:=\Xi(s_1,\ldots,s_n)}
\tag{25}
$$

**Lemma 6.** *(Asadi et al., 2020) The expected value of $\Xi$ can be bounded as:*

$$
\mathbb{E}\left[\Xi\right] \leq 2\sqrt{2}\,\mathrm{Rad}(\Phi),
\tag{26}
$$

*where $Rad$ is the Rademacher complexity.*

Given the fact that $|\Xi(s_1,\ldots,s_i,\ldots s_n) - \Xi(s_1,\ldots,s_i',\ldots s_n)| \leq \frac{2}{n}$ and apply McDiarmid's inequality, $\forall\delta\in(0,1)$, we have:

$$
\Pr\left(\Xi \leq \mathbb{E}[\Xi] + \sqrt{\frac{2\ln\frac{1}{\delta}}{n}}\right) \geq 1-\delta
\tag{27}
$$

Combining Equation 26 and Equation 27 with Equation 25, we get the following holds with probability at least $1-\delta$:

$$
\mathbb{E}_\rho\left[\left\|\pi_\beta(\cdot\mid s) - \pi_{\hat{\beta}}(\cdot\mid\phi(s))\right\|_1\right] \leq \frac{1}{n}\sum_{i=1}^n\left\|\pi_\beta(\cdot\mid s_i) - \pi_{\hat{\beta}}(\cdot\mid\phi(s_i))\right\|_1 + 2\sqrt{2}\,\mathrm{Rad}(\Phi) + \sqrt{\frac{2\ln\frac{1}{\delta}}{n}}
\tag{28}
$$

---

[7]Note that we will abuse the notation of $\pi_\phi$, $\pi(\phi(s))$, and $\pi(\cdot|\phi(s))$ for readability and simplicity.

Now from the left-hand side, with the help of Lemma 5, we get:

$$\mathbb{E}\left[J(\pi_{\hat{\beta}}) - J(\pi_{\beta})|\mathcal{D}\right] \leq \frac{2R_{\max}}{(1-\gamma)^2} \cdot \mathbb{E}\left[D_{\mathrm{TV}}\left(\pi_{\hat{\beta}}(s)\|\pi_{\beta}(s)\right)|\mathcal{D}\right]$$

$$= \frac{R_{\max}}{1-\gamma} \cdot \mathbb{E}_{\rho}\left[\left\|\pi_{\beta}(\cdot \mid s) - \pi_{\hat{\beta}}(\cdot \mid \phi(s))\right\|_1\right]. \tag{29}$$

And from the right-hand side, we have:

$$\frac{1}{n}\sum_{i=1}^{n}\left\|\pi_{\beta}(\cdot \mid s_i) - \pi_{\hat{\beta}}(\cdot \mid \phi(s_i))\right\|_1 + 2\sqrt{2}\,\mathrm{Rad}(\Phi) + \sqrt{\frac{2\ln\frac{1}{\delta}}{n}}$$

$$\leq C \cdot \frac{1}{n}\sum_{i=1}^{n}\left\|\pi_{\beta}(\cdot \mid s_i) - \pi_{\hat{\beta}}(\cdot \mid \phi(s_i))\right\|_2 + 2\sqrt{2}\,\mathrm{Rad}(\Phi) + \sqrt{\frac{2\ln\frac{1}{\delta}}{n}} \tag{30}$$

With combining Equation 29 and Equation 30 together, we can conclude the proof:

$$\epsilon_{\beta} = \mathbb{E}\left[J(\pi_{\hat{\beta}}) - J(\pi_{\beta})\,\middle|\,\mathcal{D}\right] \leq C\frac{R_{\max}}{1-\gamma} \cdot \frac{1}{n}\sum_{i=1}^{n}\left\|\pi_{\beta}(\cdot|s_i) - \pi_{\hat{\beta}}(\cdot \mid \phi(s_i))\right\|_2$$

$$+ 2\sqrt{2}\frac{R_{\max}}{1-\gamma} \cdot \mathrm{Rad}(\Phi) + \frac{R_{\max}}{1-\gamma} \cdot \sqrt{\frac{2\ln\frac{1}{\delta}}{n}} \tag{31}$$

with probability at least $1 - \delta$. $\qquad\qquad\square$

Note that the second term and the last term do not depend on the exact form of the representation. Minimizing the first term can decrease the difference between the learned policy and the behavior policy, where the first term itself aligns with the optimization problem of BPR (Equation 4), indicating that with BPR objective, we can improve the performance of the policy. And the potential harm of using BPR is bounded by the error $\epsilon_{\beta}$ that we can control.

## F   COUNTEREXAMPLE

We consider the following dataset composed of four trajectories $\tau_1, \tau_2, \tau_3$, and $\tau_4$ in deterministic MDP $m = \langle\{s_0, s_1\}, \{a_0, a_1\}, p_0(s_0) = 1, p, r, \gamma = 1\rangle$:

$$\mathcal{D} = \left\{\underbrace{\langle s_0, a_0, s_f, 0\rangle}_{\tau_1}, \underbrace{\langle s_0, a_0, s_f, 0\rangle}_{\tau_2}, \underbrace{\langle s_0, a_1, s_1, 0\rangle, \langle s_1, a_0, s_f, 1\rangle}_{\tau_3}, \underbrace{\langle s_0, a_1, s_1, 0\rangle, \langle s_1, a_1, s_f, 0\rangle}_{\tau_4}\right\},$$

where the final state $s_f$ denotes the termination of the trajectory. Then, BPR may collapse $s_0$ and $s_1$ to a single embedding $z$ where the estimated behavior policy is uniform:

$$\pi_{\hat{\beta}}(a_0|z) = \pi_{\hat{\beta}}(a_1|z) = \pi_{\hat{\beta}}(a_0|s_0) = \cdots. \tag{32}$$

Then, we have:

$$\mathcal{J}(\pi_{\hat{\beta}}) = \frac{1}{4} \tag{33}$$

$$\mathcal{J}_{\mathrm{BPR}}(\pi(a_0|z) = 1) = \frac{1}{3} \tag{34}$$

$$\mathcal{J}_{\mathrm{BPR}}(\pi(a_1|z) = 1) = 0 \tag{35}$$

$$\mathcal{J}_{\mathrm{BPR}}(\pi(a_0|z) = 0.5) = \frac{1}{2} \times \frac{1}{3} + \frac{1}{2} \times \frac{2}{3} \times \mathcal{J}_{\mathrm{BPR}}(\pi(a_0|z) = 0.5) \tag{36}$$

$$= \frac{1}{6} + \frac{1}{3} \times \mathcal{J}_{\mathrm{BPR}}(\pi(a_0|z) = 0.5) \tag{37}$$

$$= \frac{3}{2} \times \frac{1}{6} = \frac{1}{4} \tag{38}$$

As a conclusion, with a BPR value, a greedy algorithm would converge to $\pi(a_0|z) = 1$, which has the performance of 0 in the environment MDP $m$, and is therefore not a policy improvement.

---

**Algorithm 1** BPR Pretraining Pseudocode, PyTorch-like

---

```
# encoder: mlp, encoder network
# predictor: mlp, prediction network
def pretrain(encoder, predictor, replay_buffer, batch_size):
    iteration = 0
    for iteration < 1e5:
        state, action, _, _, _ = replay_buffer.sample(batch_size) # sample a batch of
            tuples from replay buffer
        state = encoder(state)
        prediction = predictor(state)
        prediction = normalize(prediction, dim=1, p=2) # l2 normalize
        action = normalize(action, dim=1, p=2) #l2 normalize
        encoder_loss = MSE(prediction, action) # loss for encoder
        encoder_loss.backward()
        update(encoder,predictor)
        iteration += 1
```

---

# G  ALL EXPERIMENTAL RESULTS

## G.1  EFFECTIVENESS OF BPR IN D4RL BENCHMARK

**Experimental Setup**   We analyze our proposed method BPR on the D4RL benchmark (Fu et al., 2020) of OpenAI gym MuJoCo tasks (Todorov et al., 2012) which includes a variety of dataset domains that have been commonly used in the Offline RL community. We evaluate our method by integrating it into three Offline RL methods, including: i)TD3+BC agent (Fujimoto & Gu, 2021), which is one of the existing state-of-the-art Offline RL approaches combining Behavior Cloning with TD3 together and has a good balancing on the simplicity and efficiency; ii) CQL agent (Kumar et al., 2020), which learns a conservative Q-function such that the expected value of a policy under this Q-function lower-bounds its true value; iii) EDAC agent (An et al., 2021), which is an uncertainty-based ensemble-diversified Offline RL algorithm. We consider three simulated tasks from the Mujoco control domain with D4RL benchmark ( halfcheetah, hopper, and walker2d) and consider the expert and medium-expert datasets. For our experiments, the baseline methods take the raw state as the input of both the value network and policy network, while we also pre-train the representations with $100k$ timesteps and then freeze the encoder and take it as the input during the Offline RL optimization; and follow the standard evaluation procedure as in D4RL benchmark. We provide the pseudocode of the pretraining process in Algorithm 1, and the training curve that without the accounts for the pre-training steps (100K timesteps) in BPR (Figure 8).

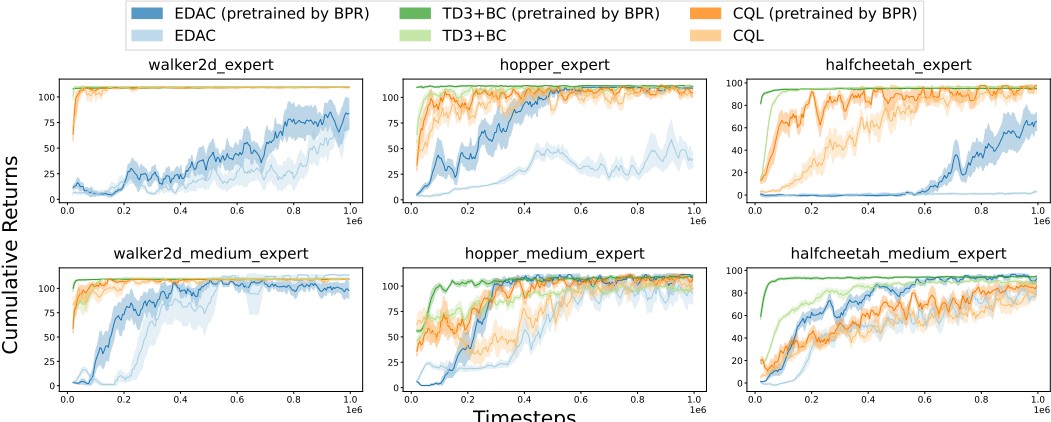

Figure 8: Performance comparison on D4RL dataset.

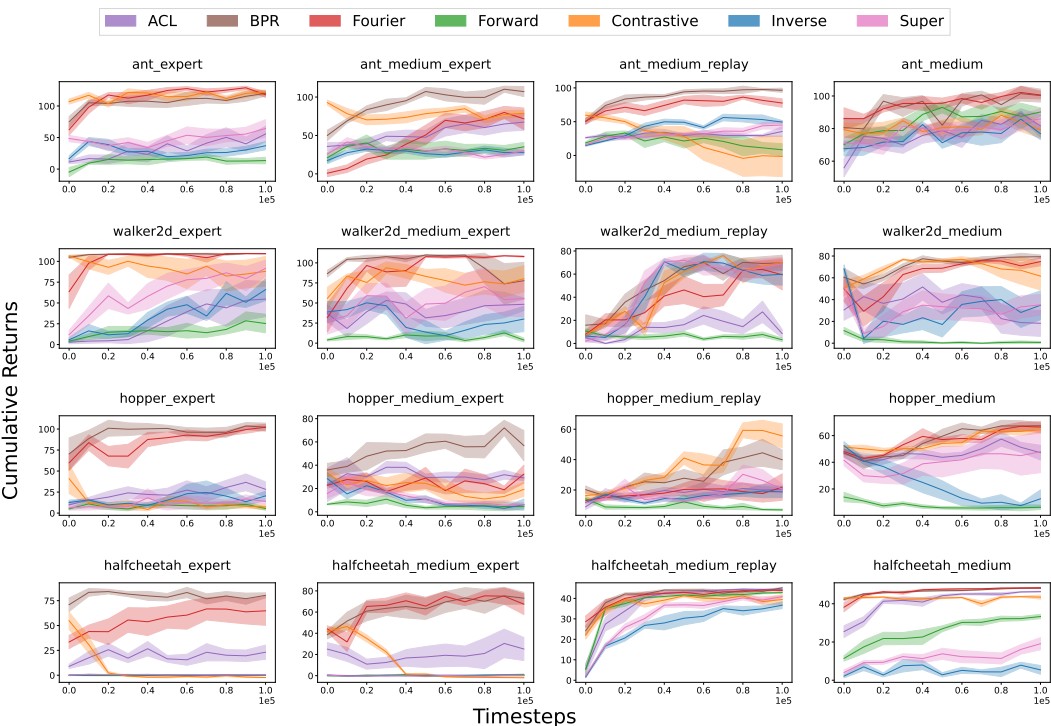

Figure 9: Performance comparison on D4RL mujoco dataset.

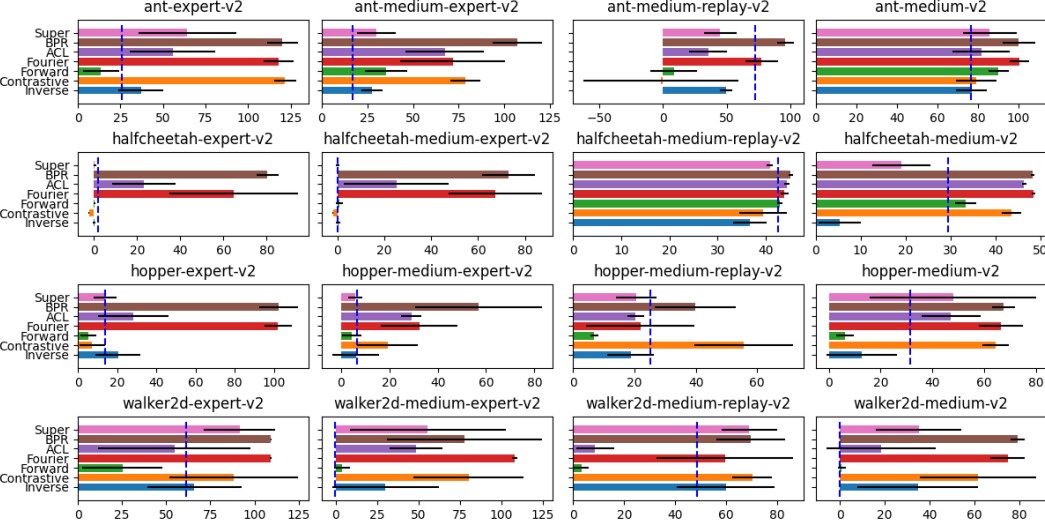

Figure 10: Performance comparison with different representation objectives on D4RL Mujoco dataset at 100K timestep. x-axis shows the average normalized score over the final 10 evaluations and 6 seeds. Blue dotted lines show the average normalized return without pretraining.

## G.2 RESULTS COMPARISON WITH OTHER REPRESENTATION BASELINES IN D4RL BENCHMARK

**Experimental Setup**    The purpose of this experiment is to demonstrate that, when compared to the state-of-the-art representation objectives, BPR still has a competitive advantage. In this experiment, we follow the pretrain-finetune paradigm suggested by Yang & Nachum (2021), i.e., we first pretrain state representations of 100k time steps, then the learned encoder is fixed and applied to the downstream task, which performs the BRAC (Wu et al., 2019) agent for 100K steps. Same as the configuration from Yang & Nachum (2021), we also fix the regularization strengths and policy learning rates in BRAC for all domains. And since representation is more valuable in informative data, our experiment does not include interaction pairs collected by a random policy, but rather has interests in the dataset collected from a learned (or at least partially learned) agent. The experiment is still conducted on the D4RL mujoco tasks, with 16 tasks in total included. In this setting, the pretraining datasets and downstream datasets are the same, determined by a single choice of task. We compare our method to several published leading representation RL algorithms, which include:

- **ACL** (Yang & Nachum, 2021) applies contrastive learning on the transformer-based architecture: (1) take a sub-trajectory $s_{t:t+k}, a_{t:t+k}, r_{t:t+k}$, (2) randomly mask a subset of these, (3) pass the masked sequence into a transformer, and then (4) for each masked input state, apply a contrastive loss between its representation $\phi(s)$ and the transformer output at its sequential position.

- **Fourier** (Nachum & Yang, 2021a) is implemented as contrastive learning where the transition dynamics are approximated by an implicit linear model with representations given by random Fourier features.

- **Forward model** (Pathak et al., 2017) uses the state representation and the action to predict the reward and next state given a sub-trajectory, where the transition probability is defined as an entropy-based model, i.e., given a sub-trajectory $\tau_{t:t+1}$, use $\phi(s_t), a_t$ to predict $s_{t+1}, r_t$.

- **Inverse model** (Pathak et al., 2017) uses the current state representation and the next state representation to predict the current action, i.e., given a sub-trajectory $\tau_{t:t+1}$, use $\phi(s_{t:t+1})$ to predict $a_t$.

- **Contrastive** (Yang & Nachum, 2021) learns state representation with contrastive objective. Given a sub-trajectory $\tau_{t:t+1}$, a contrastive loss is applied between $\phi(s_t), \phi(s_{t+1})$ as:

$$-\phi\left(s_{t+1}\right)^{\top} W \phi\left(s_t\right) + \log \mathbb{E}_{\rho}\left[\exp \left\{\phi(\tilde{s})^{\top} W \phi\left(s_t\right)\right\}\right] \qquad (39)$$

- **Super** (Yang & Nachum, 2021) is a combination of the forward model and the backward model, which has a self-predictive module and inverse module to learn dynamical information.

In this experiment, all methods use the same encoder architecture, i.e., with 4-layer MLP activated by ReLU, followed by another linear layer activated by Tanh, where the final output feature dimension of the encoder is 256. Besides, all methods follow the same optimizer settings, pre-training data, and the number of pre-training epochs.

**Analysis**    Figure 9 and Figure 10 provide the performance comparison results for BPR and other representation objectives on D4RL mujoco dataset. We find that BPR outperforms or at least is competitive with the previous state-of-the-art methods in the majority of environments. Notably, the variance on expert and medium datasets is much smaller than that on medium-expert and medium-replay datasets, especially for BPR, which indicates that the learned representation is more stable when the dataset is from the same level of the agent. The underlying reason may be that the behavior policy of the mixed dataset could differ even in the same mini-batch in the training stage, which will limit the ability of the neural network to learn the information related to (near) optimal decision-making. This suggests that the representation quality could be impacted more by the characteristics of the dataset, than the representation objectives, which confirms the observation of Schweighofer et al. (2021). We also notice that even with the pretrained encoder learned from BPR, there still remains a large performance gap between the BRAC agent and the baseline TD3+BC agent, indicating that despite the representation objective being helpful for Offline RL, choosing the suitable offline

agent for different tasks is still essential for improving the performance, which may also partially solve the aforementioned high variance issue of the different dataset. Furthermore, to understand the performance difference of different representation objectives comprehensively, we additionally use uncertainty-aware comparisons for all these methods.

## G.3 PRE-TRAINING V.S. CO-TRAINING

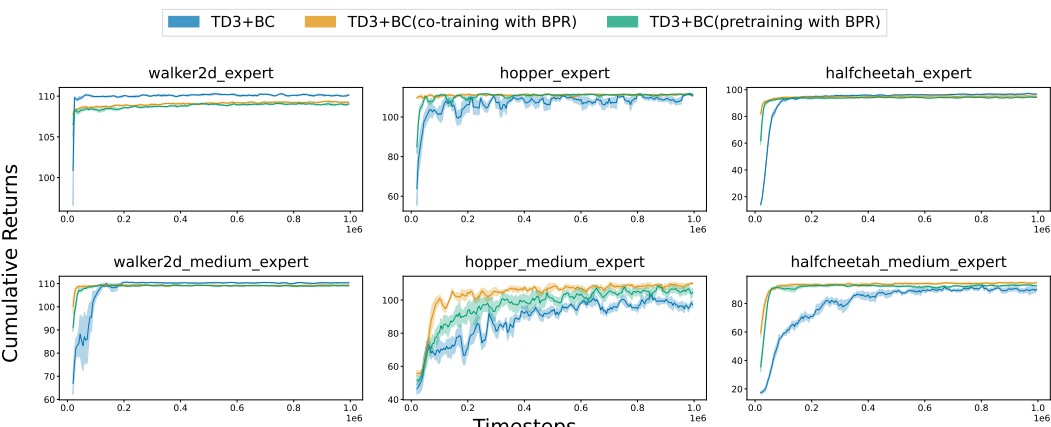

Figure 11: Performance comparison on D4RL mujoco dataset. The horizontal axis indicates the number of environment steps. The vertical axis indicates the normalized cumulative returns. We trained 6 different instances of each algorithm with different random seeds, with each instance performing an evaluation every 5,000 environment steps by computing the average episode return over 10 evaluation episodes. The solid lines represent the mean over the 6 trials.

**Experimental Setup**   There are two design choices of representation learning in the experiment: i) learn state representations via pretraining, then freeze the encoder and apply it to the downstream tasks, ii) learn state representations simultaneously with the policy, where both the representation and the policy will be learned shoulder-by-shoulder. One can wonder how BPR performs with the different designs of choices, and what the difference is between the latter one and the one that additionally applies BC to policy learning. To illustrate both questions, here, we evaluate both designs on the state-of-the-art TD3+BC backbone. For the pretraining one, we still set the pretraining stage as 100,000 time steps. For the co-training one, we train the state encoder simultaneously with the value network and the policy network, where the corresponding pseudocode is listed in Algorithm 2. For reference, we also provide the pseudocode of the baseline TD3+BC method in Algorithm 3. Concretely, the differences between BPR and the behavior cloning in TD3+BC are listed as follows: i) they use different projection (or actor) heads, ii) the gradient of the policy loss is non-visible for the encoder, but the gradient of the encoder loss can pass through and update both the encoder layer and the projection layer; iii) BPR require the l2-normalization for the representation and the action, which is known to be effective for representation learning, while $l_2$-normalization is not a good fit for action in BC.

**Analysis**   Figure 11 shows the performance of three models on D4RL tasks, indicating that BPR objective can improve sample efficiency in most cases. BPR consistently outperforms the baseline method, TD3+BC, for the majority of tasks and has a considerably lower variance, demonstrating that our approach has the advantage of convergence rate and stability. On the walker2d tasks, the baseline method performs better than the model with BPRs, we consider the possible reason could be that the baseline is sufficient to solve this task, therefore adding extra auxiliary losses may, in turn, damage the performance. Meanwhile, this finding is consistent with our hypothesis that the representation objective becomes more valuable as the difficulty of the task increases. Another impressive result is that compared to co-training BPR objective with the policy, BPR yields more performance gains and faster convergence under the pretrain-finetune paradigm, illustrating that learning policy from a "good" fixed encoder might be more suitable for Offline RL.

**Algorithm 2** TD3+BC Co-training with BPR Pseudocode, PyTorch-like

```
# actor: mlp, policy network
# critic: mlp, value network
# encoder: mlp, encoder network
# predictor: mlp, prediction network
# freq: update frequency of policy network
iteration = 0
for (s,a,ns,r) in loader: # load a minibatch tuple with n samples
    pred_s = predictor(encoder(s))
    norm_pred = normalize(pred_s, dim=1) # l2-normalize
    norm_a = normalize(a, dim=1) # l2-normalize
    L_encoder = MSE(norm_pred, norm_a) # MSE loss for encoder
    L_encoder.backward()
    update(encoder, predictor) # Adam update

    enc_s, enc_ns = sg(encoder(s)), sg(encoder(ns)) # sg is stop_gradient
    target_v = r+ gamma*sg(critic(enc_ns,a))
    L_value = MSE(critic(enc_s,a), target_v) # MSE loss for value
    L_value.backward()
    update(critic) # Adam update
    if iteration % freq == 0:
        pi = actor(enc_s)
        L_policy = -lmbda*Q(enc_s, pi).mean()+ MSE(pi,a) # loss for policy
        L_policy.backward() # back-propagate
        update(actor) # Adam update
    iteration += 1
```

**Algorithm 3** TD3+BC Pseudocode, PyTorch-like

```
# actor: mlp, policy network
# critic: mlp, value network
# freq: update frequency of policy network
iteration = 0
for (s,a,ns,r) in loader: # load a minibatch tuple with n samples
    target_v = r+ gamma*sg(critic(ns,a))
    L_value = MSE(critic(ns,a), target_v) # MSE loss for value
    L_value.backward()
    update(critic) # Adam update
    if iteration % freq == 0:
        pi = actor(ns)
        L_policy = -lmbda*Q(ns, pi).mean()+ MSE(pi,a) # loss for policy
        L_policy.backward() # back-propagate
        update(actor) # Adam update
    iteration += 1
```

## G.4 Performance comparison in v-d4rl benchmark

***Experimental Setup:*** We evaluate our method on a benchmarking suite for Offline RL from visual observations of DMControl suite (DMC) tasks (Lu et al., 2022). We consider three domains of environments (walker-walk, humanoid-walk, and cheetah-run) and two different datasets per environment (medium-expert and expert). We include two baselines from Lu et al. (2022) in this experiment:

- DrQ+BC: combining data augmentation techniques with the TD3+BC method, which applies TD3 in the offline setting with a regularizing behavioral-cloning term to the policy loss. The policy objective is: $\pi = \underset{\pi}{\arg\max}\mathbb{E}_{(s,a)\sim\mathcal{D}}\left[\lambda Q(s,\pi(s)) - (\pi(s) - a)^2\right]$

- DrQ+CQL: adding the regularization in CQL to the Q-function of an actor-critic approach, which is DrQ-v2.

To investigate if BPR learns valuable representation, we pretrain the encoder by using the BPR objective for 100k time steps, followed by a fine-tuning stage where we keep the encoder fixed and apply two baselines based on the learned encoder.

***Analysis:*** Figure 12 shows the performance of the four models on V-D4RL. We emphasize that these tasks are more challenging than their conventional D4RL counterparts. Neither DrQ+BC nor DrQ+CQL can solve them without further modifications. By integrating the frozen encoder learned from BPR, both of them obtain significant performance gains. Note that DrQ+CQL typically requires a longer training time to achieve similar performance to DrQ+BC. Nevertheless, the

substantial improvement of the modified variant of DrQ+CQL suggests that BPR can be effective in improving sample efficiency, decrease computational cost, and reach better final performance.

### G.5 PERFORMANCE COMPARISON WITH REPRESENTATION OBJECTIVES IN V-D4RL BENCHMARK

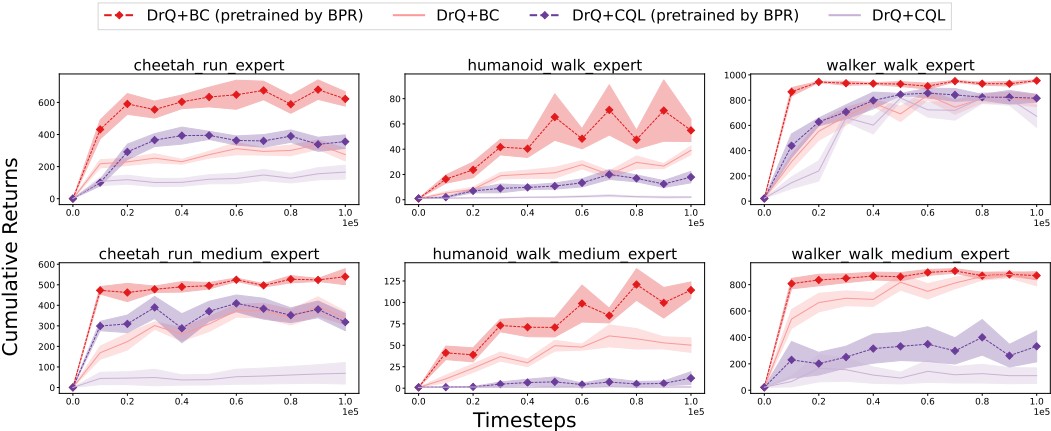

Figure 12: Performance comparison on v-d4rl benchmark, the results are averaged over 10 evaluations and 6 seeds.

**Experimental Setup**  We evaluate our method with five other representation objectives that are considered as state-of-the-art methods on a benchmarking suite for Offline RL from visual observations of DMControl suite (DMC) tasks (Lu et al., 2022). The baselines include:

- DRIML (Mazoure et al., 2020) and HOMER (Misra et al., 2020) (Time Contrastive methods) learn representations which can discriminate between adjacent observations in a rollout and pairs of random observations.

- CURL (Laskin et al., 2020) (Augmentation Contrastive method) learns a representation that is invariant to a class of data augmentations while being different across random example pairs.

- Inverse model (Pathak et al., 2017) (One-Step Inverse Models) predict the action taken conditioned on the previous and resulting observations.

- SGI (Schwarzer et al., 2021a): A combination of three kinds of techniques (Self-Predictive Representations + Inverse modelling + goal-conditioned RL). As SGI was not tested on the Deepmind Control suite (continuous-action space) setting, we modified its code[8] to allow it to be fairly compared to BPR. Specifically, for the inverse model, we follow the same architecture as the above Inverse model; for the self-predictive representation, we design a simple MLP network acting as an inverse model to capture the dynamic information; for the goal-conditioned parts, we utilize the same architecture of FiLM module in DRIML, apply DrQ+BC as the backbone, and define the potential-based reward of goal-conditioned RL to pretrain the state representations.

In this experiment, all methods use the same encoder architecture, i.e., with four convolutional layers activated by ReLU, followed by a linear layer normalized by LayerNorm  and activated by Tanh, where the final output feature dimension of the encoder is 256. Besides, all methods follow the same optimizer settings, image augmentation, pre-training data, and the number of pre-training epochs. We provide pseudocode for each method in Algorithm 4-7.

---

[8]https://github.com/mila-iqia/SGI

**Algorithm 4** DRIML(HOMER) Pretraining Pseudocode, PyTorch-like

```
# encoder: CNN, encoder network
# driml_net: mlp, contains FiLM block and residual block
def pretrain(encoder, driml_net, replay_buffer, batch_size, temp=0.1):
    iteration = 0
    for iteration < 1e5:
        state, action, state_k_step, _, _ = replay_buffer.sample(batch_size)
        # state_k_step is the k step future observation from the current state, where k=1
            in HOMER and k=5 in DRIML
        state = encoder(state)
        state_k = encoder(state_k_step)
        u_t, u_tpk = driml_net(state, state_k, action)
        outer_prod = torch.einsum('ik,jk->ij', u_t, u_tpk)
        scores = log_softmax(outer_prod / temp, -1) # temp is the temperature
        mask = torch.eye(scores.shape[-1])
        scores = (mask * scores).sum(-1).mean()
        encoder_loss = -scores # loss for encoder
        encoder_loss.backward()
        update(encoder,driml_net)
        iteration += 1
```

**Algorithm 5** CURL Pretraining Pseudocode, PyTorch-like

```
# encoder: CNN, encoder network
# W: matrix-wise parameters
def pretrain(encoder, W, replay_buffer, batch_size, temp=0.1):
    iteration = 0
    for iteration < 1e5:
        state, action, next_state, _, _ = replay_buffer.sample(batch_size)
        obs = augment(state.float()) # augment
        pos = augment(torch.clone(state).float()) # augment
        z_a = encoder(obs)
        z_pos = encoder(pos, ema=True) # using EMA in encoder network

        logits = torch.matmul(z_a, torch.matmul(W_param, z_pos.T)
        logits = logits - torch.max(logits, 1)[0][:, None]
        labels = torch.arange(logits.shape[0])
        encoder_loss = nn.CrossEntropyLoss()(logits, labels)
        encoder_loss.backward()
        update(encoder,W_param)
        iteration += 1
```

## G.6 PERFORMANCE ON THE DATASETS COLLECTED BY THE RANDOM POLICY

It should be noted that in this work, we assume the inductive bias of the behavior policy is efficient. In particular, a bad behavioral policy may lead BPR to encode an undesirable bias and therefore deteriorate the performance of the algorithm it is paired with. To verify this, we run BPR on data collected using a random policy and report the results in Figures 13 and 14. We see that the gains provided by BPR on datasets where the behavior policy is informative vanish.

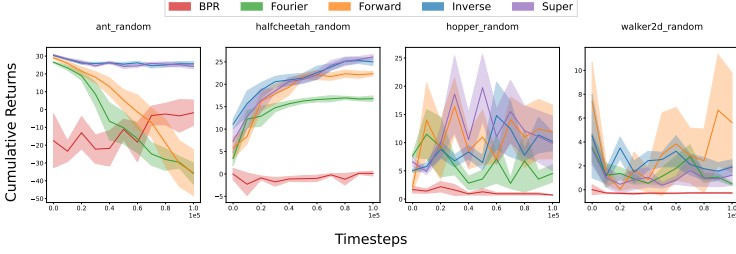

Figure 13: Performance comparison on D4RL mujoco dataset where the data is collected by a random policy.

## G.7 REPRESENTATION AS THE INPUT OF VALUE NETWORK

**Experimental Setup** As we stated in Section 5, Assumption 2 forbids in theory, the use of the BPR representation to estimate the value. In this experiment, the purpose is to show that the utilization

**Algorithm 6** Inverse model Pretraining Pseudocode, PyTorch-like

```python
# encoder: CNN, encoder network
# backward_net: mlp, inverse model
def pretrain(encoder, backward_net, replay_buffer, batch_size:
    iteration = 0
    for iteration < 1e5:
        state, action, next_state, _, _ = replay_buffer.sample(batch_size)
        state = encoder(state)
        next_state = encoder(next_state)
        state_cat = torch.cat((state, next_state), dim = -1)
        action_hat = backward_net(obs_cat)
        backward_error = torch.norm(action - action_hat, dim=-1, p=2, keepdim=True)
        encoder_loss = backward_error.mean()
        encoder_loss.backward()
        update(encoder,W_param)
        iteration += 1
```

**Algorithm 7** SGI Pretraining Pseudocode, PyTorch-like

```python
# encoder: CNN, encoder network
# predictor: mlp, predition head for forward model
# forward_net: mlp, forward model
# backward_net: mlp, inverse model
# goal_conditioner: mlp, FiLM block
# fake_actor: actor for goal-conditioned RL
# fake_critic: critic for goal-conditioned RL
def pretrain(encoder, predictor, forward_net, backward_net, goal_conditioner,
        fake_actor, fake_critic, replay_buffer, batch_size):
    iteration = 0
    for iteration < 1e5:
        state, action, next_state, _, _ = replay_buffer.sample(batch_size)
        inp_state = state
        inp_next_state = next_state

        state = encoder(state)
        next_state = target_encoder(next_state) # updated with EMA from encoder
        goal = target_encoder(goal) # updated with EMA from encoder

        sa_cat = torch.cat((state, action), dim = -1)
        next_state_hat = predictor(self.forward_net(sa_cat))
        state_cat = torch.cat((state, next_state), dim = -1)
        action_hat = backward_net(state_cat)

        forward_error = torch.norm(next_state - next_state_hat,dim=-1,p=2,keepdim=True)
        backward_error = torch.norm(action - action_hat,dim=-1,p=2,keepdim=True)
        z_g = goal_conditioner(state, goal)

        with torch.no_grad():
            goal_reward = exp_distance(next_state, goal) - exp_distance(state, goal)
            next_action = fake_actor.sample(next_state)
            target_Q1, target_Q2 = fake_critic_target(next_state, next_action)
            target_V = min(target_Q1, target_Q2)
            target_Q = goal_reward + (discount * target_V)

        Q1, Q2 = fake_critic(z_g, action)
        fake_critic_loss = MSE(Q1, target_Q) + MSE(Q2, target_Q)
        encoder_loss = fake_critic_loss + forward_error.mean() + backward_error.mean()

        encoder_loss.backward()
        update(encoder,predictor, forward_net, backward_net, goal_conditioner,fake_critic)

        state = encoder(inp_state)
        pred_action = fake_actor.sample(state)
        Q1, Q2 = fake_critic(state, pred_action)
        Q = torch.min(Q1, Q2)

        actor_policy_improvement_loss = -Q.mean()
        actor_bc_loss = F.mse_loss(pred_action, action)
        fake_actor_loss = actor_policy_improvement_loss * lambda + actor_bc_loss

        fake_actor_loss.backward()
        update(fake_actor)
        iteration += 1
```

of taking BPR representation as the input of the value network is acceptable empirically. We still pretrain the encoder and then fix it, while developing two variants of the architecture afterward: i) the learned encoder is only used for training the policy network, while the value network takes the

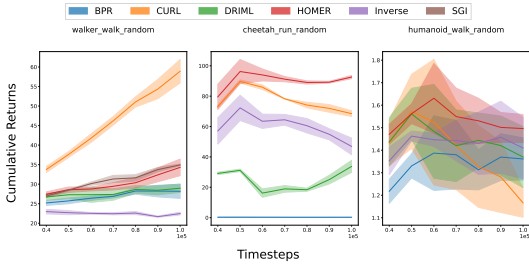

Figure 14: Performance comparison on Visual-D4RL mujoco dataset where the data is collected by a random policy.

raw state as its input; ii) the learned encoder is used as the input of both the policy network and the value network.

**Analysis** As shown in Figure 15, the performance of both variants are almost equivalent, which means that $\phi(s)$ is good enough to learn the state-action value empirically. Notably, when taking $\phi(s)$ as the input, the agent will have a lower variance along the training procedure in 2 of 6 tasks, which is the empirical evidence of that $\phi(s)$ as the input of value network is an acceptable choice.

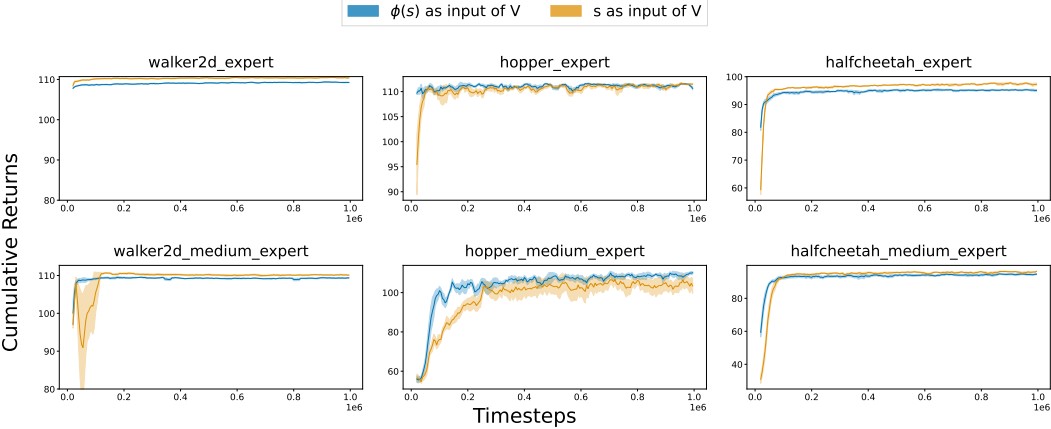

Figure 15: Performance comparison on D4RL mujoco dataset, $\phi(s)$ v.s. $s$ as input of value network.

### G.8 VISUAL D4RL BENCHMARK WITH DISTRACTORS

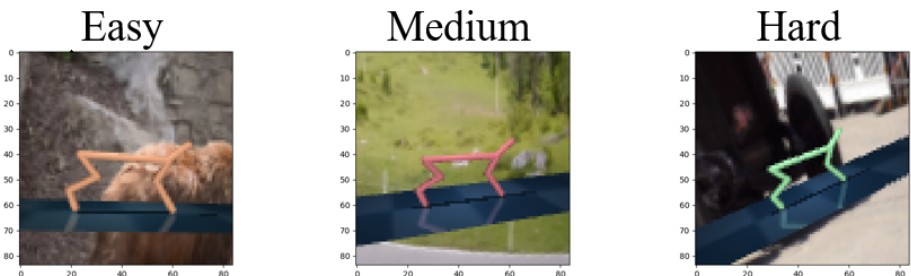

Figure 16: Pixel observations on visual d4rl benchmark with distractors.

We use three levels of distractions (i.e., easy, medium, hard) to evaluate the performance of the model (See Figure 16. Each distraction represents a shift in the data distribution, where we add task-irrelevant visual factors (i.e., backgrounds, agent colors, and camera positions) that vary with environments to disturb the model training.

