# OpenReview forum: "Behavior Prior Representation learning for Offline Reinforcement Learning"
_ICLR.cc/2023/Conference — ICLR 2023 poster_

### Official Review · Reviewer_yZFP · 2022-10-26

**Confidence:** 4
**Correctness:** 4
**Technical Novelty And Significance:** 3
**Empirical Novelty And Significance:** 4
**Recommendation:** 6

**Clarity, Quality, Novelty And Reproducibility:**

Clarity - The method and experiments are clearly explained; however, there may be a lack of clarity around the dependence on characteristics of the behavior policy.
Quality - Theoretical and experimental results are medium quality — see specific questions and comments above.
Novelty - The novelty of the method is medium. BC and offline RL have been combined in previous works (e.g., TD3+BC); however, this work proposes using BC as a state representation learning method and analyzes it from this perspective.
Reproducibility - Code is provided and datasets are public, reproducibility is high.


**Strength And Weaknesses:**

Questions and Comments

1) I am struggling to square Theorem 2 in this work with Theorem 3.4 in Li et al. 2006. If the data collection policy Beta is the optimal policy, then does a representation that satisfies Assumption 1.3 correspond to one that is \pi*-irrelevant in Li et al? I would appreciate if the authors could explain if and how the assumptions of Theorem 3.4 of Li et al. differ from their setting. The general question I think is one of representability versus learnability - it is possible to represent something but not be able to learn it.

2) It seems that there may be an implicit dependence on the characteristics of the behavior policy that is not stated. Assuming full state support does not seem to be enough - the behavior policy could visit every state but never take a good action. At the limit, suppose the behavior policy always takes the same action - then BC can achieve perfectly accuracy by completely ignoring the state input. Don’t we therefore need an assumption on the minimum return of the behavior policy? Perhaps I missed it? Theorem 5 bounds the error between the estimated and true behavior policy, but this doesn’t address the question of the worst-case error of the policy learned via offline RL with the learned state representation. In the experiments, what happens if you run the algorithm on the “random” data collection setting of D4RL?

3) I would recommend moving Figure 10 from Appendix G.5 to the main paper, as these are the main results - comparisons with other state representation learning methods on image observation inputs. What is the offline RL algorithm that is used for this experiment? What is the result here for the “random” data collection setting?

4) How is co-training with BPR different from TD3+BC? Is the difference that the BC loss is predicted with a separate head instead of being added to the TD3 loss?

5) Did you consider a probabilistic formulation for the state representation? If Z was treated as a latent random variable and the model optimized via variational inference, this would correspond to an information bottleneck on the representation Z, which would compress the information in the state representation.

Nit
 - The theorem numbering starts at 2 instead of 1

**Summary Of The Paper:**

This paper lies along the line of work exploring how to combine BC and offline RL. It takes the perspective of using BC as a state representation learning method, proposing to train an intermediate representation in the offline RL agent via BC (either as pre-training or co-training). Experimental results show that the proposed method outperforms other state representation learning methods, and offline RL from scratch, on the visual D4RL benchmark in “expert’ and “medium expert” modes.

Contributions:
Proposing BC as a state representation learning algorithm for offline RL
Theoretical analysis from a representation learning perspective
Empirical comparison to other state representation learning algorithms with visual inputs

**Summary Of The Review:**

I am concerned about the theoretical results regarding learnability of the optimal policy using this representation, and particularly how this might depend on the performance of the behavior policy. I would like to see this tested empirically as well by evaluating on the “random” data collection setting in D4RL. It is still a valuable contribution if the method only works well with certain data collection policies - however, this should be explained and tested empirically.

I am currently unconvinced about the correctness of the claims without assumptions on the behavior policy (see Questions above). This is reflected in my current scores, which I will revise upward if my concerns are addressed.


------ Update 12/9/22 --------

Thanks to the authors for the response clarifying the theoretical contributions. I also appreciate the changes the authors made to the paper to state explicitly the dependence of the quality of the learned state representation on the performance of the behavior policy, and to add experiments demonstrating this dependence.

The results in this submission are about the relative performance of the learned policy to the behavior policy, and so the sufficiency of the learned representations for representing the optimal policy (as explored in Li et al. ) is a moot point (as the authors point out).

As a side note, I do not understand the authors’ explanation: “Instead, we relax their strong requirements by allowing the value estimation in the ground MDP.” How is value estimation being done in the ground MDP when you run the offline RL algorithm in the “abstract” MDP defined by the learned state representation. In the updated appendix of the paper (page 18), it says that in contrast to Li et al., you “only change the state space,” leaving transition and reward functions unchanged. How can this be?

My concerns for the theoretical section of the paper have been addressed and I have adjusted my score for the paper accordingly.

However, I continue to think the limitation of the method to high-performing behavior policies is a significant limitation in the context of offline RL, which is often predicated on the existence of suboptimal data (indeed, when expert data is available, BC methods have been shown to be quite competitive with offline RL). In the "random" setting D4RL experiments shown in Figure 13, BPR is the worst-performing method. I also think the novelty algorithmically is rather low, consisting of a few tweaks from representation learning algorithms applied to BC. Therefore most of the contribution of the paper is carried in the empirical results that the method works better than other forms of combining BC and offline RL. I think this is a valuable enough contribution and timely given the strong interest in this topic, so I cautiously recommend acceptance.

I encourage the authors to further clarify the wording in the theoretical sections to impart the conclusions more clearly, and also to de-clutter the appendix (e.g., by removing pseudo-code) to make it easier to find results (such as the "random" D4RL experiment illustrating a failure case of the method).

---

> ### Author Response · Authors · 2022-11-14
> **Response to Reviewer yZFP (1/2)**
>
> We thank the reviewer for the detailed feedback on the manuscript. As per the comments, we have indeed updated the draft and included additional experimental results to demonstrate the suitability of BPR and more detailed comparisons with the related approaches to illustrate the advantage of BPR. Please see the generic response made to all reviewers to address this.  Our answer for each of the questions raised in detail is as follows:
>
> **Q1: The connection between Theorem 2 in this work with Theorem 3.4 in Li et al. 2006**
>
> **Answer:** Thank you for the reference, the biggest difference is that: we assume that we use the abstraction only for the policy search (as stated in Assumption 2.2), while [1] applies $\pi^*$-irrelevance abstraction to both value estimation and policy updates in $\bar{M}$. Therefore, [1] requires much stronger assumptions for their policy abstractions, such as bisimulation or invariance for the optimal value/policy. Instead, we relax their strong requirements by allowing the value estimation in the ground MDP [^1]. By doing so, we can perform policy optimization on the abstract MDP with the guarantees enunciated in Section 5 in the manuscript. Note that we do not have any guarantees on the near-optimality in the true environment. However, we argue that this is not an issue as the policy optimization problem in the Offline RL setting is intractable under an arbitrary behavioral policy [2,3]. Also, we do not address directly the learnability of the optimal policy in the abstract state space, as such guarantees are limited with neural networks, but they should be the same as with the learnability of the optimal policy in the full state space. We have also updated the main submission with a paragraph about this discussion in Appendix C.2.
>
>
> **Q2: Implicit dependence of the behavior policy**
>
> **Answer:** In this work, we assume the inductive bias of the behavior policy is efficient, in such a way that where BPR can leverage this information which was ignored before in the literature. As stated in the No Free Lunch Theorem, a bad behavioral policy may lead BPR to encode an undesired bias and therefore deteriorate the performance of the algorithm it is paired with. However, since it is common that we know whether or not a dataset has been collected with an expert policy or a bad policy, either beforehand or by simply observing the dataset's empirical return, such information can be used to decide whether or not to use BPR.
>
> **Q3: “...suppose the behavior policy always takes the same action - then BC can achieve perfectly accuracy by completely ignoring the state input”**
>
> **Answer:**
> - BPR will fail to improve in this particular setting, but we want to first point out that such a degenerate dataset does not contain any information to infer any kind of policy improvement. So BPR will not help but not be harmful either. A more adversarial setting is the uniform policy in every state and this is what we discuss and test below.
> - if the policy is random, then BPR's representation is expected to be uninformative, and even worse, to be losing important information to guide efficient action taking. As a result, BPR's representation is expected to hinder the policy improvement and the algorithm to return a policy that has the same performance as the behavior, with no improvement (which is its worst case). To validate this hypothesis, we have run experiments on the random dataset, and find that as expected, BPR suffers in that setting (see Figures 13 and 14).
> - However, we argue that the suitability of BPR can be decided beforehand. As an example, we can use trajectory quality [4] as a measurement where we only apply BPR when the trajectory quality is above a certain threshold. And our empirical results on non-random datasets show that it works very well in practice and requires a very low runtime.
>
> ---
>
> [^1]: *We have done the corresponding experiments in G.7, and we found that the performance difference between the ground state as the input and the abstraction as the input is trivial, showing that estimating value in representation space is also acceptable empirically*
>
> ### References
>
> [1]: Lihong Li, Thomas J. Walsh, Michael L. Littman: Towards a Unified Theory of State Abstraction for MDPs. AI&M 2006
>
> [2]: Chenjun Xiao, Ilbin Lee, Bo Dai, Dale Schuurmans, Csaba Szepesvári: The Curse of Passive Data Collection in Batch Reinforcement Learning. AISTATS 2022: 8413-8438
>
> [3]: Dylan J. Foster, Akshay Krishnamurthy, David Simchi-Levi, Yunzong Xu: Offline Reinforcement Learning: Fundamental Barriers for Value Function Approximation. COLT 2022: 3489
>
> [4]: Kajetan Schweighofer, Markus Hofmarcher, Marius-Constantin Dinu, Philipp Renz, Angela Bitto-Nemling, Vihang P. Patil, Sepp Hochreiter: Understanding the Effects of Dataset Characteristics on Offline Reinforcement Learning. CoRR abs/2111.04714 (2021)

---

> ### Author Response · Authors · 2022-11-14
> **Response to Reviewer yZFP (2/2)**
>
> **Q4: worst-case error to the optimal**
>
> **Answer:** We emphasize that training the optimal policy from the fixed dataset is intractable in general [1,2] and therefore our method does not aim at achieving it. Instead, we use an inductive bias that we prove to be efficient for many datasets. And the worst-case scenario of BPR is that the policy has no performance gain from the learned representations, which means it falls back to the estimated behavior (as we explained below Theorem 2), so that the largest gap is the optimal value minus the behavior value (and the approximation errors).
>
> **Q5: moving Figure 10 from Appendix G.5 to the main paper; describing the backbone algorithm in Figure 10**
>
> **Answer:** Thanks for the valuable suggestions. We have updated the manuscript by replacing the comparison to baseline RL algorithms with the comparison to other representation objectives and providing a description of the backbone algorithm now.
>
>
> **Q6: How is co-training with BPR different from TD3+BC?**
>
> **Answer:** We provide the corresponding pseudocode of TD3+BC with BPR objective and the baseline TD3+BC algorithm in Algorithm 2,3. The differences between BPR and the behavior cloning in TD3+BC are: i) they use different projection (or actor) heads, ii) for BPR, the gradient of the policy loss is non-visible for the encoder, but the gradient of the encoder loss can pass through and update both the encoder layer and the projection layer; iii) BPR uses the $l_2$-normalization for the representation and the action, which is known to be effective for representation learning, while $l_2$-normalization is not a good fit for action in BC.
>
> **Q7: Probabilistic formulation for the state representation (information bottleneck)**
>
> **Answer:** Thanks for the good suggestion! We agree with the reviewer that if we treat the state representation as a latent random variable, we can draw a connection with state representation with an information bottleneck, which can balance the trade-off between the ability of the latent space to compress (rate) and represent the original signal (distortion). It is an interesting direction, adding some form of the bottleneck on the latent space is definitely worth exploring in future work.
>
> **Q8: The theorem numbering should start from 2 instead of 1**
>
> **Answer:** Thanks! We have revised the number in the new manuscript.
>
> ### References
>
> [1]:  Chenjun Xiao, Ilbin Lee, Bo Dai, Dale Schuurmans, Csaba Szepesvári: The Curse of Passive Data Collection in Batch Reinforcement Learning. AISTATS 2022: 8413-8438
>
> [2]: Dylan J. Foster, Akshay Krishnamurthy, David Simchi-Levi, Yunzong Xu: Offline Reinforcement Learning: Fundamental Barriers for Value Function Approximation. COLT 2022: 3489

---

> ### Author Response · Authors · 2022-11-28
> **Kindly Reminder**
>
> Dear reviewer,
>
> Thanks again for the time spent on our manuscript and your valuable feedback. We have revised the paper to address your questions and comments. Hopefully, the current version of the paper addresses your initial concerns.
>
> Did you have the opportunity to read our response? Is there something else you would like us to clarify or address? We are also looking forward to your feedback on our reply. Please let us know if you have any additional comments.

---

> ### Author Response · Authors · 2022-12-12
> **Thank you for updating the scores**
>
> Dear Reviewer,
>
> Thank you very much for updating the scores for our paper and providing detailed feedback. We appreciate your comments and suggestions, which have helped us improve the paper significantly. In response to your additional concerns:
>
> - We meant that value estimation is performed on the natural input (which belongs to the ground MDP state space), while the paper by Li et al. performs it on the representation space, which raises issues that can only be alleviated with their drastic assumptions. We will revise the descriptions accordingly for the camera-ready version. Note that this is an observation for the theoretical part of the paper. We find in Appendix G.7, Figure 15 that, in practice, using the representation for value estimation too works as well if not better.
>
> - We agree that the limitation of our method to high-performing behavior policies is a significant limitation in the context of offline RL. However, we want to stress that it is easy to use information about the behavior policy (either prior knowledge or statistics on the returns in the dataset) to decide when to apply BPR. Additionally, there are various approaches that can be used to improve the quality of the datasets, such as trajectory stitching [1,2] and data re-balancing [3,4]. These approaches can help to improve the effectiveness of BPR when applied to randomly sampled datasets. As a potential extension, we could explore combining some of these approaches with BPR to further improve policy performance.
>
> - We agree that the theoretical sections of our paper could be clearer, and we will make sure to revise the wording to make our conclusions more evident. We will also review the appendix and make any necessary changes to improve its organization and clarity.
> Thank you again for your valuable feedback. If you have any further comments or suggestions, please let us know.
>
> Sincerely,
>
> Authors
>
>
> ### References
>
> [1]: Charles A. Hepburn, Giovanni Montana: Model-based Trajectory Stitching for Improved Offline Reinforcement Learning. CoRR abs/2211.11603 (2022)
>
> [2]: Ian Char, Viraj Mehta, Adam Villaflor, John M. Dolan, Jeff Schneider: BATS: Best Action Trajectory Stitching. CoRR abs/2204.12026 (2022)
>
> [3]: Yang Yue, Bingyi Kang, Xiao Ma, Zhongwen Xu, Gao Huang, Shuicheng Yan: Boosting Offline Reinforcement Learning via Data Rebalancing. CoRR abs/2210.09241 (2022)
>
> [4] Anonymous: Harnessing Mixed Offline Reinforcement Learning Datasets via Trajectory Weighting. Submitted to ICLR2023. https://openreview.net/forum?id=OhUAblg27z

---

### Official Review · Reviewer_vYat · 2022-10-28

**Confidence:** 4
**Correctness:** 4
**Technical Novelty And Significance:** 2
**Empirical Novelty And Significance:** 2
**Recommendation:** 6

**Clarity, Quality, Novelty And Reproducibility:**

Clarity: I found the paper clear and well structured

Quality: High quality, the clearly a lot of work went into this paper. The theoretical analysis in particular appears to be sound.

Novelty: I am not well versed enough in offline representation learning for RL to assess the novelty. Though I would like to bring to the authors attention the work of Le Lan et al. 2022 which also defines a notion of effective dimension in the context of representation learning.

Reproducibility: I have not gone through the supplementary material in enough depth to comment on reproducibility. The information given in the main text would not be enough to reproduce the results.


[1] Le Lan et al., "On the Generalization of Representations in Reinforcement Learning" 2022

**Strength And Weaknesses:**

Strengths:

* The work addresses an important open problem in offline RL.
* Very well motivated
* The presentation is clear and coherent making it a pleasant read
* Some interesting theoretical guarantees are provided
* The proposed method is evaluated on different criteria (performance against baseline algorithms without pretrained representations, comparison against different representation objectives, change in the effective dimension of the data, and robustness to distractors in pixel-based inputs)

Weaknesses:

* The theoretical analysis provides some intuition on the kind of performance gain is expected when the representation is trained on a data under a policy $\pi_{\beta}$. This analysis essentially means that the quality of the $\pi_{BPR}$ highly depends on the behaviour policy. Authors should discuss address what to do in cases where the behavioural policy is highly adversarial for instance. What does that mean for the learned representation? Is it still beneficial to learn a BPR, or is it harmful then?

* When assessing the effectiveness of BPR against baselines (non-pretrained representation), a fairer comparison would account for the additional 100k training steps BPR used for pretraining. I understand the argument that the representation needed not to be retrained for all of the different offline RL methods, still it feels like an unfair comparison.

* When comparing against other against other representation objectives, a couple things need clarifications:
  * How were the baseline selected? (the performance criterion used is from Schwarzer et al. 2021, however the SGI representation is not included in the baselines?)
  * Are the encoders for the different objectives of similar architecture/capacity?

* I would have like to see a discussion or some results comparing BRP with representations that embed the transition dynamics (e.g. self predictive representations)

**Summary Of The Paper:**

The work focuses on learning representation for offline reinforcement learning. The authors propose a representation learning method based on cloning the behaviour under which the offline dataset was collected. The main motivations of the work is to (1) reduce the difficulty of learning a policy from limited high-dimensional data by projecting the observations into a representative low-dimensional space, and (2) combat the under-parameterization problem of value networks that arises when bootstrapping is used by disentangling the representation learning from policy/value learning.

The proposed representation learning method is independent of the policy learning RL algorithm and can thus be used with any offline RL algorithm. Authors provide theoretical policy improvement guarantees under some assumption and present empirical results, showing outperformance against baselines.

**Summary Of The Review:**

Interesting, well motivated and well presented paper. However the experiments are is not detailed enough/choices made are not justified clearly enough. No intuition is given for addressing the limitation of the work as it stands. For these reasons my score is 5.

---

> ### Author Response · Authors · 2022-11-14
> **Response to Reviewer vYat (1/2)**
>
> Thank you for your review and favorable assessment of our work. As per the comments, we have indeed updated the draft and included additional experimental results to demonstrate the suitability of BPR and more detailed comparisons with the related approaches to illustrate the advantage of BPR. Please see the generic response made to all reviewers to address this. We provide a response to your comments below:
>
> **Q1: discuss when the behavioural policy is highly adversarial for instance**
>
> **Answer:** The question is related to similar questions 1 and 2 from the reviewer yZFP. The main risk is when some information in the state is ignored by the behavioral policy. Then, the BPR representation may/will ignore this information and will not be able to make any improvements. In order to show this effect, we consider the extreme situation, with the least informative behavioral policy: the uniformly random policy. In this case, using BPR degrades the Offline RL algorithm (see Figure 13 and Figure 14). This being said, we would like to recall that, in this work, we assume that the inductive bias of the behavior policy is high performing, meaning that the actions taken by the behavioral policy carry relevant information that BPR is the first to leverage in the Offline RL literature. We would also like to stress that it is common that we know whether or not a dataset has been collected with an expert policy or a bad policy, either beforehand or by simply observing the dataset's empirical return. Such information can be used to decide whether or not to use BPR.
>
> **Q2: Provide a fairer comparison (account for the additional 100k training steps BPR used for pretraining)**
>
> **Answer:** Thanks, we have added curves that account for the pre-training steps in Figure 2 in the main draft as per your suggestion, and the previous Figure 2 is moved to Figure 8 in Appendix G.1 now (and will be removed eventually). Notably, we had followed the exact 100K pre-training steps for learning state representation used [1,2,3], where these approaches show the performance only by considering the training steps that start from the actual policy learning since the state representation learning can be disentangled from policy learning. However, our results show that whether the pre-training steps are considered or not, BPR still outperforms baselines quite rapidly.
>
>
> **Q3: SGI [4] is not included in the baselines**
>
> **Answer:** The implementation of SGI was specifically designed for Atari100K tasks, and the authors had not provided the code for the continuous-action tasks, this is the reason why we did not include it as a baseline before. However, we agree that SGI is a state-of-the-art representation method that should be compared. Therefore, we have modified the official code the authors provided and adapted it to the continuous-action tasks. We provide the details of the modification in Appendix G.5. After being evaluated with the same criteria, the empirical results for SGI are now included in the main text (Figure 4) and show that our approach still outperforms SGI in most of the tasks in terms of policy performance. Furthermore, in terms of the run time comparison, our approach is 2.5x faster than SGI when evaluating the wall-clock time of pretraining for 100K time steps. These empirical results illustrate our approach still outperforms the other representation objectives.
>
> ### References
>
> [1]: Mengjiao Yang, Ofir Nachum: Representation Matters: Offline Pretraining for Sequential Decision Making. ICML 2021: 11784-11794
>
> [2]: Ofir Nachum, Mengjiao Yang: Provable Representation Learning for Imitation with Contrastive Fourier Features. NeurIPS 2021: 30100-30112
>
> [3]: Cynthia Chen, Xin Chen, Sam Toyer, Cody Wild, Scott Emmons, Ian Fischer, Kuang-Huei Lee, Neel Alex, Steven H. Wang, Ping Luo, Stuart Russell, Pieter Abbeel, Rohin Shah: An Empirical Investigation of Representation Learning for Imitation. NeurIPS Datasets and Benchmarks 2021
>
> [4]: Max Schwarzer, Nitarshan Rajkumar, Michael Noukhovitch, Ankesh Anand, Laurent Charlin, R. Devon Hjelm, Philip Bachman, Aaron C. Courville: Pretraining Representations for Data-Efficient Reinforcement Learning. NeurIPS 2021: 12686-12699

---

> ### Author Response · Authors · 2022-11-14
> **Response to Reviewer vYat (2/2)**
>
> **Q4: Are the encoders for the different objectives of similar architecture/capacity?**
>
> **Answer:** All encoders for the different representation objectives are the same for a fair comparison. Moreover, all methods follow the same optimizer settings, pre-training data, and the number of pre-training epochs. We have added more details in Appendix G.2 and G.5.
>
> **Q5: Comparison to self-predictive representations**
>
> **Answer:** Thanks for the suggestion. We had already included a Forward model as a baseline in the d4rl experiment at submission, and apologize for the lack of description. The Forward model baseline learns self-predictive representations: its inputs are the state representation $\phi(s)$ and action $a$, its output is the next state representation $\phi(s’)$, and it aims to minimize the reconstruction loss. For further comparison with predictive models, we additionally include the Super model [1] as a baseline, which has both forward and inverse modules. Our representation objective still surpasses these baselines. (For details of the models please refer to Appendix G.2).
>
> **Q6: Effective dimension related work**
>
> **Answer:** Thanks for bringing this work to our sight! The effective dimension used in this paper is inspired by the Feature rank defined in [2]. As suggested, we have added a comparison with the Effective Dimension proposed by [3] in Appendix D. In summary, both measurements are derived from the Singular Value Decomposition (SVD), while one ([2]) is derived from the singular value of the Gram matrix, and the other one ([3])  is derived from the left singular vector. Considering the state representation from this perspective would be a potential future work, thanks again!
>
> **Q7: The information given in the main text would not be enough to reproduce the results**
>
> **Answer:** We admit that the descriptions of the baselines may not have been enough for a comprehensive comparison. In the updated manuscript, we have added more details about our objective and the other representation baselines in Appendix G.2 and Appendix G.5 (the space limit does not allow us to include this information in the main text). We also provided the pseudocode of the representation baselines that were used in the visual-d4rl datasets, and further extended the Reproducibility Statement section for the readers to reimplement all of the experiments. On acceptance, the code will be publicly released (we provide the code in supplementary materials for now).
>
> ### References
>
> [1]: Mengjiao Yang, Ofir Nachum: Representation Matters: Offline Pretraining for Sequential Decision Making. ICML 2021: 11784-11794
>
> [2]: Clare Lyle, Mark Rowland, Will Dabney: Understanding and Preventing Capacity Loss in Reinforcement Learning. ICLR 2022
>
> [3]: Charline Le Lan, Stephen Tu, Adam Oberman, Rishabh Agarwal, Marc G. Bellemare: On the Generalization of Representations in Reinforcement Learning. AISTATS 2022: 4132-4157

---

> > ### Comment · Reviewer_vYat · 2022-11-21
> > **Thanks for addressing my questions**
> >
> > Thanks for addressing all my questions and comments and including clarifications in the updated manuscript. I've adjusted my score accordingly.

---

> > > ### Author Response · Authors · 2022-11-22
> > > **Thank you for updating the scores**
> > >
> > > Dear Reviewer,
> > >
> > > Thank you very much for updating the scores for the paper. We appreciate your detailed feedback and comments on our work, which helped improve the paper significantly.
> > >
> > > If you have any further feedback on the draft, please let us know too.
> > >
> > > Thanks.

---

### Official Review · Reviewer_jmKm · 2022-11-01

**Confidence:** 3
**Correctness:** 4
**Technical Novelty And Significance:** 3
**Empirical Novelty And Significance:** 4
**Recommendation:** 8

**Clarity, Quality, Novelty And Reproducibility:**

The quality of the work is high given the amount, clarity, and significance of the experimental results. I do not qualify to give a proper evaluation of the entirety of the theoretical work.

**Strength And Weaknesses:**

Strengths:
- Clear motivation with plenty of references giving the unfamiliar reader the necessary context
- A large number of experiments showing significant performance gain
- Ablation studies to investigate Robustness to Visual Distractions and Effective Dimension
- Substantial effort to provide theoretical guarantees

Weaknesses:
- A few parts of the paper seem rushed (in contrast to the remaining part of the paper). E.g. the wordings in the Reproducibility Statement, "?other representation objectives?", and it seems to me that state and action are swapped in Figure 1 under BPR.

**Summary Of The Paper:**

This paper presents a new pre-training method called Behavior Prior Representation (BPR) for offline RL that applies Behavioral Cloning with a decoder that learns a compact state representation that is then used by offline RL methods. This compact representation is shown to alleviate the implicit under-parameterization phenomenon that causes over-aliasing of features. The paper presents theoretical performance guarantees and demonstrates significant performance gains with BPR on the D4RL Offline RL benchmarks.

**Summary Of The Review:**

This paper is of high quality, has a clear motivation, and a simple and well-founded approach that is evaluated with a substantial amount of experimental and theoretical work.

---

> ### Author Response · Authors · 2022-11-14
> **Response to Reviewer jmKm**
>
> Thank you for your review! We’re glad you appreciated our work. We have revised our manuscript following your comments. Specifically:
> -  We apologize for the mistake we made in Figure 1, we have corrected it in the updated manuscripts.
> - We have added more details about our objective and the other representation baselines in Appendix G.2 and Appendix G.5.
> - We also provided the pseudocode of the representation baselines that were used in the visual-d4rl datasets, and further extended the Reproducibility Statement section for the readers to reimplement all of the experiments. On acceptance, the code will be publicly released (we provide the code in supplementary materials for now).
> - To further improve the quality of our work, we have indeed updated the draft and included additional experimental results to demonstrate the suitability of BPR and more detailed comparisons with the related approaches to illustrate the advantage of BPR. Please see the generic response made to all reviewers to address this.
>
> We hope to have addressed all your questions and concerns. We would also be grateful if you can let us know if there are any other concerns that we can address further. Thank you for your help and time.

---

### Author Response · Authors · 2022-11-14
**General response to all reviewers: The suitability of BPR**

We sincerely thank the reviewers for the insightful comments about the limit case and the suitability of BPR. We are taking account of this, and going forward, would include more details to emphasize its adaptability to different scenarios, which we omitted previously for sake of brevity. For clarification to reviewers, we summarize the suitability of BPR here, the additional details are provided at the end of the manuscript (please see the updated draft).

- Optimality in Offline RL has been recently shown to be intractable, even under sufficient coverage of the behavior [1,2], this suggests that inductive biases are required to train good policies in a reasonable amount of time.

- We explore behavior cloning as an inductive bias for representation learning (as opposed to standard policy optimization), and show empirically that it works really well.

- However, a consequence of this objective is that the only guarantee we can provide theoretically is an improvement over the behavior policy (Theorems 2 and 3), not optimality. This type of theoretical guarantee is also a common practice in the Offline RL literature [3,4,5].

- Indeed, if the policy is adversarial or random, then BPR may return a policy that has the same performance as the behavior, but no improvement (which is the worst case). In order to challenge the limits of BPR, we have run experiments on the random dataset, and find that as expected, BPR's representation is a liability to the policy optimization in that setting (Figures 13 and 14).

- We can leverage information about the behavior (either a priori or from statistics on the returns in the dataset) to decide when to use BPR. As an example, we can use trajectory quality [6] as a measurement where we only apply BPR when the trajectory quality is above a certain threshold. And our empirical results on non-random datasets show that it works very well in practice and that our solution is computationally efficient.

- We have added a discussion on this in the main text, and the random dataset results in the appendix.

### References

[1]: Chenjun Xiao, Ilbin Lee, Bo Dai, Dale Schuurmans, Csaba Szepesvári: The Curse of Passive Data Collection in Batch Reinforcement Learning. AISTATS 2022: 8413-8438

[2]: Dylan J. Foster, Akshay Krishnamurthy, David Simchi-Levi, Yunzong Xu: Offline Reinforcement Learning: Fundamental Barriers for Value Function Approximation. COLT 2022: 3489

[3]: Thomas, P. S. Safe reinforcement learning. PhD thesis, University of Massachusetts Libraries, 2015.

[4]: Mohammad Ghavamzadeh, Marek Petrik, Yinlam Chow: Safe Policy Improvement by Minimizing Robust Baseline Regret. NIPS 2016: 2298-2306

[5]: Romain Laroche, Paul Trichelair, Remi Tachet des Combes: Safe Policy Improvement with Baseline Bootstrapping. ICML 2019: 3652-3661

[6]: Kajetan Schweighofer, Markus Hofmarcher, Marius-Constantin Dinu, Philipp Renz, Angela Bitto-Nemling, Vihang P. Patil, Sepp Hochreiter: Understanding the Effects of Dataset Characteristics on Offline Reinforcement Learning. CoRR abs/2111.04714 (2021)

---

### Author Response · Authors · 2022-11-14
**General response to all reviewers: Updated draft**

Based on all the reviewers' helpful feedback, we have made numerous changes to the paper including adding clarifying sections and new experiments in the appendix (**all important changes are colorized blue in the manuscript**). We hope this revision will further help the reviewers reach a consensus on the value of our work and foster further discussion.

## Major updates in the manuscript

###  Additional theoretical results

1. We have added a detailed comparison with the $\pi^\star$-irrelevance abstraction introduced by Li et al (2006) [1] in Appendix C.2. In [1], a $\pi^*$-irrelevance abstraction attempts to preserve the optimal action when aggregating states (Definition 3.5 in [1]), and is applied to both value estimation and policy updates in an abstract MDP $\bar{M}$ (Definition 1 in [1]). Therefore, [1] requires much stronger assumptions for the policy abstractions, such as bisimulation or invariance for the optimal value/policy (Theorem 3 in [1]). In comparison, we instead use the abstraction for policy search only, thus we can perform policy optimization with the guarantees enunciated in the submission.

2. The effective dimension used in this paper is inspired by the Feature rank defined in [2]. As suggested, we have added a comparison with the Effective Dimension proposed by [3] in Appendix D. In summary, both measurements are derived from the Singular Value Decomposition (SVD), while one ([2]) is derived from the singular value of the Gram matrix, and the other one ([3])  is derived from the left singular vector.

### Additional empirical results

1. For the offline RL tasks with visual-based observations, i.e., visual-d4rl dataset, we have added SGI (from [4]) as a new baseline, We also added run time comparisons for several representation objectives(in Figure 4). In this experiment, our approach still outperforms SGI in most tasks in terms of policy performance, furthermore, it is 2.5x faster than SGI when evaluating the wall-clock time of pretraining for 100K time steps, which shows the effectiveness of our work.

2. For the offline RL tasks with raw-state, i.e., d4rl dataset, we added the Super Model (from [5]) as a new baseline (in Figure 3, Table 1, Figure 9, and Figure 10). The Super Model combines a self-predictive objective with an inverse RL one. BPR surpasses Super Model in most cases, and the performance gain is still statistically significant.

3. We shifted the x-axis for BPR algorithms in Figure 2 to account for the 100k pretraining steps. BPR still outperforms baselines quite rapidly.

4. For the results on the random dataset, we have added the performance plot for both visual and raw-state input settings (Figure 13 and Figure 14) and the corresponding analysis in Appendix G.6.

### Additional descriptions

1. Detailed descriptions of each representation baseline (including the objectives, the encoder architectures, and some pseudocode)

2. Detailed descriptions in the reproducibility statement; Additional limitations in the Discussion section.

3. The backbone offline RL algorithm for the experiment of the performance comparison of representation objectives on visual observation datasets (Figure 10 previously, Figure 4 now).

4. The difference between TD3+BC co-training with BPR and the TD3+BC baseline (provided the pseudocode along with the description).

## Minor changes in the manuscript

1. Moved Figure 10 (in the previous version of the manuscript) to the main manuscript as Figure 4 now, and put the previous Figure 4 (in the previous version of the manuscript) to the appendix. We thank Reviewer yZFP for the suggestion.
2. Updated the numbering of the theorems (previously we used the same counter for the theorems and the definitions). We thank Reviewer yZFP for pointing this out.
3. Updated Figure 1 to meet specifically with the description of BPR; updated the description in the Reproducibility Statement section: “plug BPR in any other Offline RL algorithms” -> “plug BPR in any Offline RL algorithms”. We thank Reviewer jmKm for pointing this out.


### References

[1]: Lihong Li, Thomas J. Walsh, Michael L. Littman: Towards a Unified Theory of State Abstraction for MDPs. AI&M 2006

[2]: Clare Lyle, Mark Rowland, Will Dabney: Understanding and Preventing Capacity Loss in Reinforcement Learning. ICLR 2022

[3]: Charline Le Lan, Stephen Tu, Adam Oberman, Rishabh Agarwal, Marc G. Bellemare: On the Generalization of Representations in Reinforcement Learning. AISTATS 2022: 4132-4157

[4]: Max Schwarzer, Nitarshan Rajkumar, Michael Noukhovitch, Ankesh Anand, Laurent Charlin, R. Devon Hjelm, Philip Bachman, Aaron C. Courville: Pretraining Representations for Data-Efficient Reinforcement Learning. NeurIPS 2021: 12686-12699

[5]: Thomas, P. S. Safe reinforcement learning. PhD thesis, University of Massachusetts Libraries, 2015.

---

### Author Response · Authors · 2022-11-15
**Summary of Responses and Updated Manuscript**

We thank all the reviewers for their time and their acknowledgment of our work (such as being well-motivated, substantial effort to provide theoretical guarantees, and comprehensive evaluations on different criteria). More importantly, we thank all the reviewers
for their consideration in providing useful feedback for this manuscript. We hope our responses would help them re-evaluate their scores, as we tried to address all their concerns. We have updated our draft with new experiment results (including additional baselines and the performance on the random datasets) and new theoretical results (the connection and the difference between BPR with related approaches such as $\pi^*$-irrelevance abstraction), meanwhile we have also additionally provided the limit case of our approach. This should hopefully provide better insights into the suitability of BPR. Furthermore, we have also addressed each of the reviewer's questions individually as responses here. We sincerely hope this would help reviewers understand this work better, and help them re-evaluate the scores for the paper.

---

### Author Response · Authors · 2022-11-18
**General Response**

Dear reviewers and AC:

We thank all the reviewers for their valuable  feedback. We have revised the paper to address questions and comments, and we also have tried to provide responses to all questions. With a gentle reminder, we, as the authors, will not be allowed to further revise the draft after November 18. We would like to address any further questions, if any. We hope our responses and updated manuscript with new results would help you further re-evaluate and improve the score of the paper.

Best,

Authors

---

### Decision · Program_Chairs · 2023-01-20

**Decision:**

Accept: poster

**Justification For Why Not Higher Score:**

While this is a clearly a good paper, the idea is not novel enough (imho) for a spotlight presentation.

**Justification For Why Not Lower Score:**

The paper is well-written, the results are good, and there is sufficient novelty to accept the paper.

**Metareview: Summary, Strengths And Weaknesses:**

The idea behind this contribution can be summed up easily (which is a strength): pre-train using behavioral cloning, and then use the learned representations to do offline RL. The paper shows both theoretically and empirically that this works well and gives appreciable benefits to standard offline RL methods. I and the reviewers concur that this is good work with some general applicability.

**Note From Pc:**

if the above contains the word "oral" or "spotlight" please see: "oral" presentation means -> notable-top-5% and "spotlight" means -> notable-top-25%. As stated in our emails, we are disassociating presentation type from AC recommendations